# The oldest three-dimensionally preserved vertebrate neurocranium

Richard P. Dearden[1,2 ✉], Agnese Lanzetti[1,3], Sam Giles[1,3], Zerina Johanson[3], Andy S. Jones[1], Stephan Lautenschlager[1], Emma Randle[1] & Ivan J. Sansom[1]

The neurocranium is an integral part of the vertebrate head, itself a major evolutionary innovation[1,2]. However, its early history remains poorly understood, with great dissimilarity in form between the two living vertebrate groups: gnathostomes (jawed vertebrates) and cyclostomes (hagfishes and lampreys)[2,3]. The 100 Myr gap separating the Cambrian appearance of vertebrates[4–6] from the earliest three-dimensionally preserved vertebrate neurocrania[7] further obscures the origins of modern states. Here we use computed tomography to describe the cranial anatomy of an Ordovician stem-group gnathostome: *Eriptychius americanus* from the Harding Sandstone of Colorado, USA[8]. A fossilized head of *Eriptychius* preserves a symmetrical set of cartilages that we interpret as the preorbital neurocranium, enclosing the fronts of laterally placed orbits, terminally located mouth, olfactory bulbs and pineal organ. This suggests that, in the earliest gnathostomes, the neurocranium filled out the space between the dermal skeleton and brain, like in galeaspids, osteostracans and placoderms and unlike in cyclostomes[2]. However, these cartilages are not fused into a single neurocranial unit, suggesting that this is a derived gnathostome trait. *Eriptychius* fills a major temporal and phylogenetic gap in our understanding of the evolution of the gnathostome head, revealing a neurocranium with an anatomy unlike that of any previously described vertebrate.

Efforts to understand the evolution of the vertebrate head have been hampered by the strikingly different anatomies encountered in cyclostomes and gnathostomes. In living cyclostomes, the neurocranium comprises an open framework of cartilages holding the brain and a feeding apparatus consisting of a symmetrical set of paired and midline cartilages[9,10]. In gnathostomes, the brain is instead enclosed by a single neurocranial unit that encases the brain and nasal capsules, with paired mandibular and hyoid arches forming the feeding apparatus[11]. In the absence of a suitable extant outgroup, the fossil record is crucial to polarize cyclostome and gnathostome character states and reveals character states in extinct taxa that can be used to test transformational hypotheses[2,7,12,13].

However, the fossil record does little to bridge the two states, with major phylogenetic and temporal sampling gaps in early vertebrate cranial anatomy. Three-dimensional remains of galeaspids, osteostracans and placoderms, the Silurian and Devonian stem-group gnathostomes most closely related to the crown group, suggest that the plesiomorphic state for gnathostomes is a single, endocranial unit formed by the neurocranium and splanchnocranium, which encloses the brain and pharynx and fills the connective tissue compartments between them and the dermal skeleton[1,7,14–16]. However, in the multiple other groups of Palaeozoic vertebrates all that is known of the cranial anatomy comes from difficult-to-interpret two-dimensional fossils[4,6,17,18] or is inferred on the basis of the dermal skeleton[19–21]. Although flattened remains exist of the crania of some Lower Cambrian vertebrates[4,6], an expanse of time separates these from the Silurian galeaspids preserving three-dimensional neurocrania[7,22,23]. Of those few Ordovician vertebrate taxa that are known from dermal remains[24], nothing is known of the endocranial anatomy.

The enigmatic *Eriptychius americanus*, from the Sandbian (458.4–453.0 million years ago) Harding Sandstone, is one of only four Ordovician stem-group gnathostome taxa known from articulated remains[25–35] and the only one in which putative cartilages have been identified[8]. In this study, we use computed tomographic methods to image an articulated specimen of *E. americanus*, PF 1795, and confirm that these cartilages represent the earliest-known three-dimensionally articulated neurocranium in a fossil vertebrate. The cranial cartilages of *Eriptychius* are anatomically dissimilar from the crania of both cyclostomes and gnathostomes and instead represent a distinct pattern of the vertebrate head skeleton.

## Systematic palaeontology

Subphylum: Vertebrata Lamarck, 1801
Eriptychius *americanus* Walcott, 1892
*Eriptychius americanus* Walcott, p. 167, plate 4, figures 5–11

**Type material.** Seven isolated fragments of dermal bone embedded in matrix, collected by C. D. Walcott from the Harding Quarry, Cañon City,

[1]School of Geography, Earth & Environmental Sciences, University of Birmingham, Birmingham, UK. [2]Naturalis Biodiversity Centre, Leiden, The Netherlands. [3]Natural History Museum, London, UK. ✉e-mail: r.dearden@bham.ac.uk

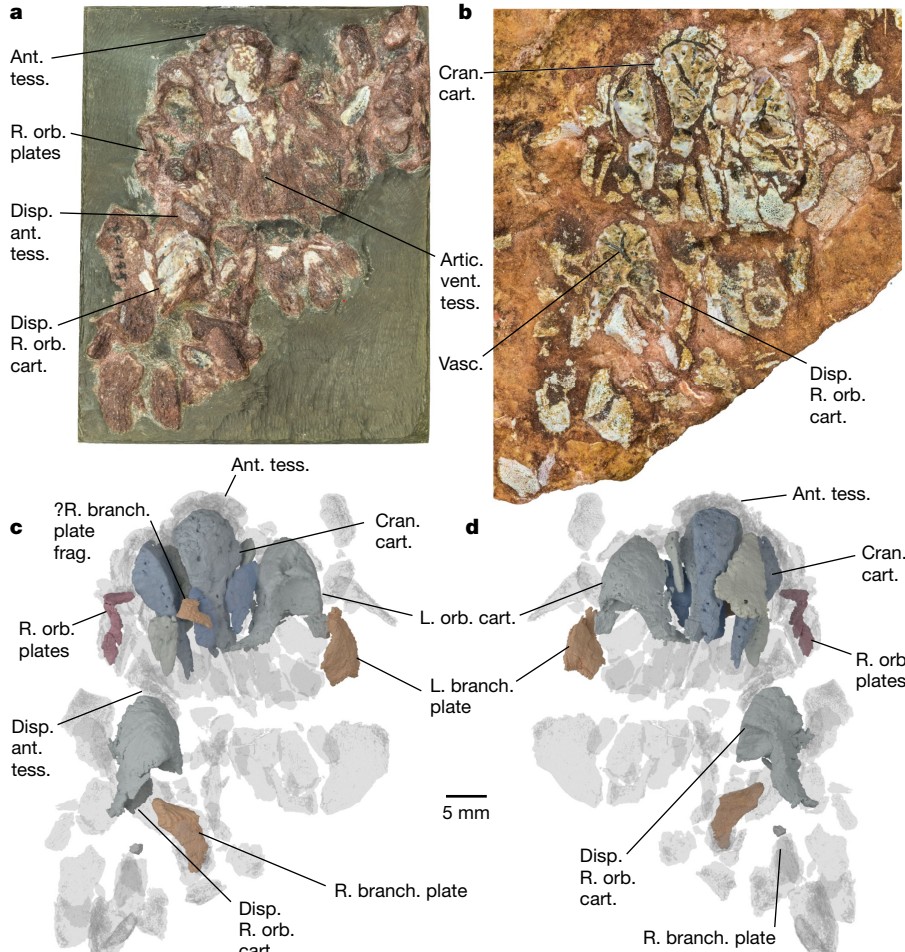

**Fig. 1 | Overview of *Eriptychius americanus* PF 1795. a,b**, Photographs of part PF 1795a, which had the split face set in epoxy and was manually prepared (**a**), and its counterpart PF 1795b, which remains in rock matrix (**b**). Both are shown in an anatomically ventral view. **c,d**, Digital model of computed tomographic data of the combined part and counterpart with most of dermal skeleton rendered transparent: anatomical ventral view (corresponding to the visible area of the part in epoxy) (**c**) and anatomical dorsal view (buried in matrix in the counterpart) (**d**). Colour scheme for renders: blue-greys, cranial cartilages (matching the detailed scheme in Fig. 2); transparencies, the dermal skeleton; orange, branchial plates; red, orbital plates. Anterior to top in **a**–**d**. ant. tess., anterior tesserae; artic. vent. tess., articulated ventral tesserae; branch. plate, branchial plate; cran. cart., cranial cartilages; disp., displaced; frag., fragment; L., left; orb. cart., orbital cartilage; orb. plates, orbital plates; R., right; vasc., vasculature; ?, probable. Scale bar applies to all panels.

Colorado, form the syntype USNM V 2350 in the collections of the Smithsonian National Museum of Natural History in Washington, DC, USA.

**Emended diagnosis.** Agnathan with mesomeric dermal tesserae and scales formed from acellular bone overlain by ornament formed from coarse wide-calibre tubular dentine. Body scales covered in multiple elongate ridges. Antorbital neurocranium comprising symmetrical set of elements containing numerous large canals internally, endoskeleton closely associated with but not fused to the surrounding dermal skeleton. Shared with *Astraspis*, arandaspids, other 'ostracoderms' excluding heterostracans: multiple branchial openings. Shared with *Astraspis* and tessellate heterostracans: dermal head skeleton formed from dorsal and ventral 'headshields' of ornamented tesserae. Differs from *Astraspis* in that ornament of dermal skeleton comprises ridges and presence of coarse tubular dentine.

**Description.** Computed tomography scanning of the part and counterpart of PF 1795 (here termed PF 1795a and b, respectively; Methods) confirms the identity of this material as a partially articulated *Eriptychius* head[8], including components of both the dermal skeleton and endoskeleton (Fig. 1, Extended Data Figs. 1–3 and Supplementary Video; see Supplementary Information for comments on histology). The articulated individual is confined to the near surface of the matrix;

below it is a mash of isolated elements typical of the Harding Sandstone bone beds including additional tesserae referable to *Eriptychius* that do not seem to be associated with the articulated specimen (Extended Data Fig. 1). Denison[8] described several large elements of globular calcified cartilage as part of the internal skeleton of *Eriptychius* and we have been able to distinguish ten separate cartilages in total comprising the endoskeletal cranium (Figs. 1 and 2 and Extended Data Figs. 2 and 4).

Six cartilages were identified on the split surface of PF 1795a by ref. 8 (Extended Data Fig. 2) and the remaining four are buried within the matrix of PF 1795b. The cartilages are closely wrapped by articulated squamation anteriorly, dorsally and ventrally and to one side (Extended Data Figs. 3 and 4); however, there is a clear separation between dermal and endoskeletal elements, unlike in galeaspids and osteostracans[36]. This squamation comprises a range of dermal element types including the scale types identified by ref. 8 and scales with an anteroposteriorly oriented ornament from farther back on the head. It also includes small, curved orbital plates (Fig. 1c,d and Extended Data Fig. 3c–e) and several plates similar in morphology (Fig. 1c,d and Extended Data Fig. 3f–i) to an isolated *Eriptychius* 'branchio-cornual plate' identified by ref. 37 (plate 2, 4–7). There is no obvious pineal dermal plate.

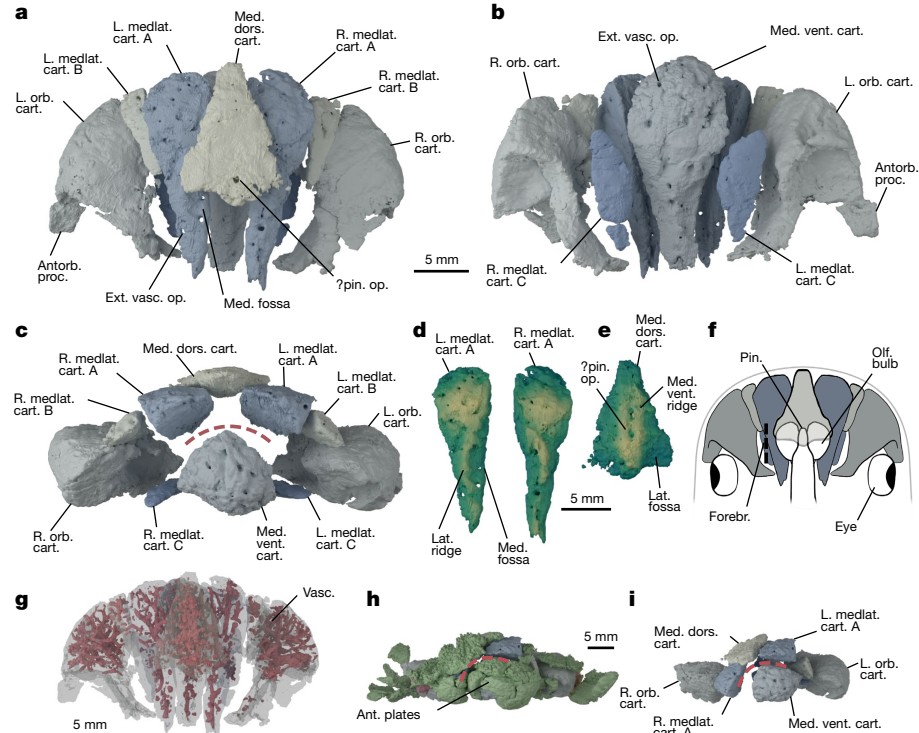

**Fig. 2 | The neurocranial cartilages of *E. americanus* PF 1795 based on computed tomography data. a–c**, Cranial cartilages in estimated life position, with cartilages coloured in pairs in dorsal (**a**), ventral (**b**) and anterior (**c**) view. **d**,**e**, Mediolateral cartilages A in dorsal view (**d**) and median dorsal cartilage in ventral view (**e**) rendered with a vertical height map texture to emphasize the surface topology. **f**, Reconstruction of the forebrain relative to the cranial cartilages using a lamprey as a model[9,52], shown in dorsal view. **g**, Cartilages in dorsal view, rendered transparent to show internal vasculature (red). **h**,**i**, Cartilages in preserved position in anterior view with dermal skeleton shown (**h**) and removed (**i**). Colours in **a**,**b**,**c**,**f**,**h**,**i** as in Fig. 1 with the following additions. Green, dermal skeleton. Red dashed line represents inferred position of mouth in **c**,**h**,**i**. In **d** and **e** lighter colours denote areas closer to the camera. Abbreviations as in Fig. 1 with the following additions: antorb. proc., antorbital process; ext. vasc. op., external vascular openings; forebr., forebrain; lat., lateral; medlat. cart, mediolateral cartilage; med. dors. cart, median dorsal cartilage; med., medial; med. vent. cart., median ventral cartilage; med. vent. ridge, median ventral ridge; olf. bulb, olfactory bulb; pin., pineal organ; pin. op., pineal opening; vent., ventral. Scale bar in **a** is shared by **b**,**c**; scale bar in **d** is shared by **e**.

Denison identified two cartilages as the orbital cartilages on the basis of concave posterior faces[8]. Our scan data confirm that these represent cross-sections through large fossae that we interpret as forming the anterior wall of the orbits. These fossae are flanked laterally by an antorbital process (Fig. 2a,b and Extended Data Fig. 5) that suggests a dorsolateral orientation of the orbit. A smaller ventral fossa on each orbital cartilage below the orbit may have provided a location for muscle attachment (Fig. 2). One orbital cartilage is posteriorly displaced, along with elements of anterior squamation including a rostral scale (Fig. 1 and Extended Data Fig. 3) and has rotated by about 180° in the sagittal axis; when rotated back into position it aligns with the orbital plates (Fig. 1). The anteriormost branchial plate lies posterolateral to the other orbital cartilage (Fig. 1 and Extended Data Fig. 3), suggesting that the relative positions of the orbits, otic region and branchial arches were similar to those inferred in heterostracans[2,19], although this is impossible to judge exactly because of the collapse of the dermal skeleton. The absence of anything assignable to the branchial skeleton suggests that the branchial arches were not incorporated into a single mineralization with the neurocranium, a major difference with respect to galeaspids and osteostracans[7,15].

The remaining cartilages lie between the two orbital cartilages, although they have slumped slightly and been pulled posteriorly on one side with the displaced orbital cartilage (Figs. 1 and 2a–c and Extended Data Fig. 4). There are three paired cartilages arranged symmetrically across the midline—two dorsal (termed mediolateral A, B) and one ventral (termed mediolateral C)—along with one unpaired midline cartilage dorsally and one ventrally. Of the two unpaired cartilages, the smaller dorsal element is kite-shaped and has both the dorsal and ventral surfaces punctured by large medial foramina that we consider likely to be the pineal opening (Fig. 2a,e). These dorsal and ventral foramina do not exactly line up anteroposteriorly but appear to communicate with a common large space inside the cartilage (Fig. 2g and Extended Data Fig. 6). This kite-shaped cartilage is preserved overlying the mediolateral cartilages A and its ventral surface bears a median ridge with a shallow depression on either side (Fig. 2a,e). Together with shallow depressions on the dorsal surface of mediolateral cartilages A, these depressions frame paired fossae which we interpret as having accommodated part of the forebrain, possibly the olfactory bulbs (Fig. 2d–f), an interpretation consistent with their position relative to the orbits and putative pineal opening. Thus, we infer this to be the dorsal side of the animal and term this the median dorsal cartilage. The larger median ventral cartilage underlies the mediolateral cartilages.

All cartilages are pervasively penetrated by canals (Fig. 2g and Extended Data Fig. 6). In the larger cartilages, that is, the orbital cartilages and the mediolateral cartilages A, this tends to follow the pattern of a larger trunk entering the cartilage from the posterior side before splitting into smaller branches that open to the surface. The pattern is not exactly bilaterally symmetrical in the paired and unpaired elements but does follow a similar organization with the trunk canal entering at equivalent points. These canals could plausibly have carried sensory rami onto the surface of the head; for example, the superficial

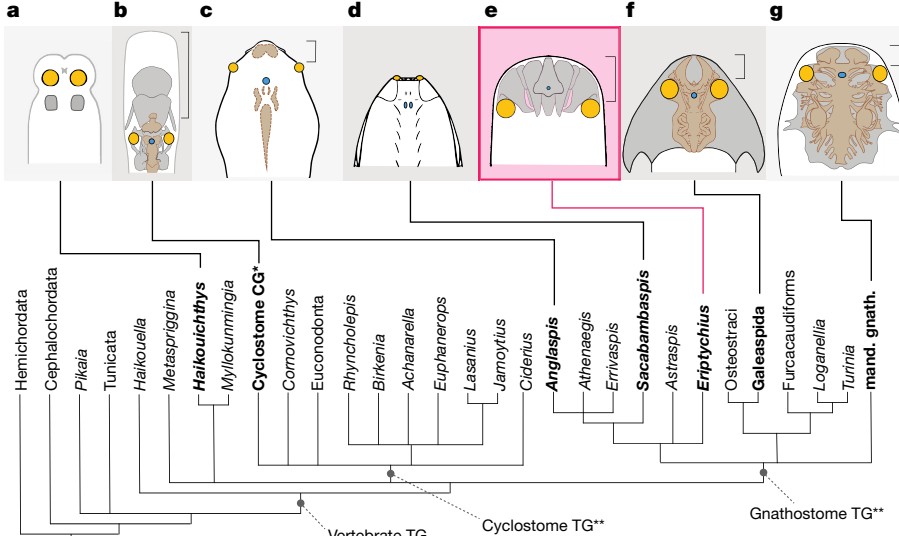

**Fig. 3 | The neurocrania of vertebrates in a phylogenetic framework.**
**a**, *Haikouichthys*, adapted from ref. 3, CC-BY 4.0. **b**, *Petromyzon*. **c**, Generalized cyathaspid, adapted from ref. 20, Springer Nature. **d**, *Sacabambaspis*, reproduced from ref. 31, copyright ©1993 Elsevier Masson SAS, all rights reserved (but see ref. 53 for alternative interpretation). **e**, *Eriptychius*. **f**, *Shuyu*, adapted from ref. 7, Springer Nature. **g**, *Dicksonosteus* adapted with permission from ref. 14, Muséum national d'histoire naturelle, Paris. Phylogenetic scheme is the Adams consensus tree described in the text, bold taxon names represent those depicted above. Coloured regions illustrate positions of key sensory organs: yellow, eyes; blue, pineal. Light grey lines represent body outlines, black lines represent dermal body armour, light grey regions with dark grey borders represent cranial mineralizations, brown regions represent the brain (**b**), imprints of the brain in the dorsal headshield (**c**) or the endocast (**f**,**g**). Square brackets represent the preorbital part of the head. *Petromyzon* redrawn from Sketchfab (https://sketchfab.com/3d-models/lamprey-inside-c9f05e3f00c94d929e7ed018fac6d782) under a CC BY 4.0 licence. Abbreviations: CG, crown group; mand. gnath., mandibulate gnathostomes; TG, total group. *'Crown-group cyclostomes' represents the Petromyzontidae and Myxinoidea total groups and *Gilpichthys*, which was recovered in a polytomy with those two groups. **Cyclostome and gnathostome total groups in this topology recovered in a polytomy with *Metaspriggina* and (*Haikouichthys* + *Myllokunmingia*).

ophthalmic nerve in the case of the canals in the orbital cartilages. However, the canal openings are not restricted to surfaces where the cartilages contact the dermal skeleton. On the basis of this and the lack of sensory canal openings in the head tesserae, they may have carried vasculature instead or as well. In living chondrichthyans, canals carry vasculature and transport prechondrocytes into the cartilage from the perichondral surface. This could implicate the canals in *Eriptychius* in both cartilage maintenance and interstitial cartilage growth, although in modern chondrichthyans the width of these canals are an order of magnitude smaller than in *Eriptychius*[38,39]. Although canals have not been explicitly reported in other early vertebrate cartilages, sections through galeaspid cartilage suggest that this tissue has a degree of vascularity[36].

In concert with the displacement of the cartilages, the dermal squamation has undergone postmortem collapse. We interpret the articulated area of squamation as having covered the right side of the head, around the region of the right orbit and the right side of the mouth, as well as the areas dorsal and ventral to the mouth (Fig. 1 and Extended Data Fig. 3). On the basis of this interpretation, the cartilages would have comprised the preorbital region of the head, surrounding the mouth (Fig. 2c,h,i). The articulated dorsal and ventral patches of squamation indicate that the mouth opened between the cartilages, bordered by the 'rostral' plates[8] and was oriented supraterminally (Fig. 2c,h,i), unlike most jawless stem-gnathostomes (notable exceptions being *Doryaspis*[40] and *Drepanaspis*[41]). The fact that the cartilages are separate may mean that some movement of the oral endoskeleton was possible, although this would have presumably been limited by their close relationship with the squamation.

## The evolution of vertebrate crania

The cranial cartilages of *Eriptychius* have no obvious homologue in the head of any known extant or extinct vertebrate (Fig. 2 and Extended Data Fig. 4). The most obvious comparison is with the numerous paired and midline cartilages that comprise the complex feeding apparatus of cyclostomes[42,43] and possibly anaspid-like early vertebrates[17]. The preorbital region of *Eriptychius* is ostensibly similar in that it comprises numerous cartilages that bordered the mouth. Unlike cyclostomes, however, in which the entire brain is held in an open cartilaginous framework[10], in *Eriptychius* the forebrain at least, as well as the orbits, were bounded by these cartilages. The only large cartilage that is not associated with the forebrain or orbit, the median ventral cartilage, is closely associated with the dermal skeleton, suggesting that mobility would have been restricted. Although it is impossible to rule out movement of the smaller mediolateral cartilages B and C, a cyclostome-like mobile feeding apparatus seems unlikely.

The preserved cranial cartilages of *Eriptychius* instead seem to have formed a static structure, which, in this sense, is more comparable to neurocrania in known jawless stem-group gnathostomes. In these taxa, osteostracans and galeaspids, endoskeletal tissues fill the connective tissue space between the dermal skeleton and the brain, where they surround the mouth and pharynx and buttress the head[1,7,14]. The cartilages in *Eriptychius* are similar in that they fill out the head and closely support the dermal skeleton. Unlike osteostracans and galeaspids, however, the neurocranial cartilages are separate from one another and there is no evidence for any mineralization from the level of the orbits posteriorly or for the fusion of the splanchnocranium into a unit with the neurocranium. This could be taphonomic but parts of the dermal skeleton posterior to the orbits remain articulated in PF 1795. Instead, *Eriptychius* may have resembled heterostracans, in which the otic region of the brain and the pharynx are closely associated with the dermal skull roof[19,44], suggesting that they were primarily supported by the dermal skeleton. The patterning of developmental cartilages in extant gnathostomes might suggest that the mineralization of this anterior region in *Eriptychius* was limited to the prechordal region of

the neurocranium (the trabeculae cranii), which originate from neural crest[3,12]. However, it has been demonstrated that in lampreys the parachordals extend forward to form most of this region[3,12], another possible origin for the cartilages in *Eriptychius*.

*Eriptychius* fills an important gap, both temporal and phylogenetic, in our understanding of the evolution of the vertebrate head. Our inclusion of *Eriptychius* in a recent phylogenetic matrix for early vertebrates[45,46] recovers it within or in a polytomy with the vertebrate crown group and as a stem-group gnathostome in the Adams consensus of the parsimony analysis (Fig. 3 and Extended Data Figs. 7–9) which is consistent with previous phylogenetic analysis[35]. This phylogenetic placement would extend the condition in which endoskeletal tissues fill space in the neurocranium into the earliest, Ordovician, stem-group gnathostomes and show that this is not limited to taxa with an osteostracan/galeaspid morphology of a ventrally positioned mouth and dorsally located orbits. However, the fact that in *Eriptychius* the cartilages are not fused into a single unit around the brain suggests that these early gnathostome neurocrania calcified in several parts. An enclosing neurocranium, which in galeaspids and osteostracans is fused with the splanchnocranium, is a trait which unites galeaspids, osteostracans and mandibulate gnathostomes to the exclusion of *Eriptychius* and cyclostomes. Although it is possible to identify broad similarities, the substantial difference between the neurocranial anatomy of the Ordovician *Eriptychius* to either cyclostomes or gnathostomes helps to explain why it has been so difficult to draw direct anatomical comparison between the skulls of the two living vertebrate groups[2].

*Eriptychius* provides the earliest direct evidence for a prechordal endocranium in a vertebrate. This was also likely to have been the case in the contemporary *Astraspis* and the later heterostracans based on the lateral positions of the orbits[33,47,48] (Fig. 3d,e). This contrasts with *Sacabambaspis*, in which the orbits are placed at the extreme anterior margin of the headshield[30] (Fig. 3c) comparable to those of putative stem-group vertebrates[4] such as *Haikouichthys, Metaspriggina* and conodonts[4–6,49] (Fig. 3a) as well as in the naked anaspids *Jamoytius* and *Euphanerops*[50,51]. In the past, this unusual anatomy has usually been dismissed as a specialization on the basis of interpretations of *Sacabambaspis* in a heterostracan light (for example, refs. 27,31 and supplementary appendix (p35) of ref. 45). The discovery of this preorbital neurocranium in *Eriptychius* and the movement of early vertebrate taxa around the vertebrate crown node in recent phylogenies[45,46] should prompt reconsideration of whether differences in orbital placement in Ordovician vertebrates instead reflect a more fundamental anatomical difference in cranial organization.

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

## Methods

### Specimen

*E. americanus* PF 1795 is held at the Field Museum, Chicago, USA. It comprises two pieces, part and counterpart, one of which had the split face set in epoxy and was manually prepared out of the matrix from the other side[8]. Denison[8] figured only the part in epoxy (before and after preparation) and referred to both part and counterpart as PF 1795: here, we term the part in epoxy PF 1795a and the unprepared counterpart PF 1795b for the sake of clarity (Fig. 1).

### Geological setting

The Harding Sandstone (Sandbian, Upper Ordovician) is a thin unit of interbedded mudstones and sandstones that extends around the Cañon City embayment in the frontal range of the Rockies. The sequence was first studied in detail by ref. 54 focusing on the presence of Ordovician vertebrates with periodic attention paid subsequently. Sedimentologically, the sequence records shallow marine deposition within a microtidal lagoonal setting with localized estuarine input[55]. The extensive bone beds that occur through the Harding represent winnowed shoreface deposition with specimen PF 1795 described here thought to come from an equivalent or potentially the same horizon to the articulated specimen of *Astraspis desiderata* that represents a shoreface strandline[33].

### Scanning

Computed tomography was carried out at the University of Chicago on a Phoenix v|tome|x with a dual 180 tube. PF 1795a was scanned at a voltage of 100 kV and current of 370 µA with a 0.1 mm Cu filter, for 1,800 projections, achieving a voxel size of 34.17 µm. PF 1795b in the matrix was scanned at a voltage of 110 kV and current of 300 µA with no filter, for 2,000 projections, achieving a voxel size of 44.8940 µm. Both datasets were segmented in Mimics v.25 (materialize) to create three-dimensional meshes using manual segmentation with some interpolatory functions ('3D interpolate' and 'Multiple Slice Edit'). These were exported in the PLY format and then visualized in Blender (blender.org) v.3.3.0. An additional, higher resolution scan of PF 1795a was carried out at the University of Bristol in an effort to better visualize the endoskeletal tissue; this was carried out at a voltage of 120 kV and current of 119 µA with no filter obtaining 3,141 projections with a voxel size of 14.72 µm.

### Phylogenetic analysis

The phylogenetic analysis was conducted on the basis of the matrix of ref. 45, with minor modifications focussed on Ordovician vertebrates. *Eriptychius* was added, astraspids recoded as *Astraspis* and arandaspids as *Sacabambaspis*. We also revised the codings of heterostracan taxa, revising *Athenaegis* and splitting Heterostraci into *Anglaspis* and *Errivaspis*. Changes are listed and justified in the Supplementary Information. This resulted in a matrix including 54 taxa and 167 morphological characters, which we analysed using parsimony and Bayesian analyses.

Parsimony analysis was carried out in TNT v.1.5 using a parsimony ratchet and TBR branch swapping with 10,000 replicates, holding 100 trees from each iteration, with the constraint (Hemichordata (Cephalochordata (Tunicata + all other taxa))) and Hemichordata set as the outgroup. This resulted in 1,951 equally parsimonious trees with a length of 351. Bayesian analysis was carried out in MrBayes v.3.2.7, a flat (uniform) prior was used with an Mkv model and a gamma-distributed rate parameter. Hemichordata was set as the outgroup and total-group vertebrates were constrained to be monophyletic. We carried out the search for 10,000,000 generations, sampling a tree every 1,000 generations and calculated a majority rule consensus tree with a relative burn-in of 25%.

### Reporting summary

Further information on research design is available in the Nature Portfolio Reporting Summary linked to this article.

## Data availability

The computed tomography scan data this work is based on are available via Morphosource via the following links: PF 1795a (https://doi.org/10.17602/M2/M510644) and PF 1795b (https://doi.org/10.17602/M2/M510665). A mesh incorporating all three-dimensional surface models generated in this study is made available via Morphosource at https://doi.org/10.17602/M2/M542071. Individual three-dimensional meshes and the phylogenetic matrix are made available on figshare at https://doi.org/10.6084/m9.figshare.23726487.

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

**Acknowledgements** Many thanks to B. Simpson and A. Stroup (FMNH) for providing specimen access and loan and to A. Neander (University of Chicago) and L. Martin-Silverstone (University of Bristol) for help with computed tomography scanning the specimen. We also thank D. Elliott and M. Coates for constructive discussion over the years and logistic help with the specimen. Thanks to L. Schnetz for assistance with Adobe Illustrator. This work was supported by the Leverhulme Trust project 'Feeding without jaws—innovations in early vertebrates'. R.P.D. is at present supported by the Marie Skłodowska-Curie Action 'DEADSharks'. Thanks to the Willi Hennig Society for making the program TNT available.

**Author contributions** This study was conceived by R.P.D. and I.J.S. Additional computed tomography scanning was carried out by S.G. Segmentation of the data, analysis and visualization in figures were carried out by R.P.D., with discussion with all other authors. The initial version of the manuscript was written by R.P.D.; it was then developed with input from all authors. All authors accepted the final version of the manuscript.

**Competing interests** The authors declare no competing interests.

**Additional information**
**Correspondence and requests for materials** should be addressed to Richard P. Dearden.

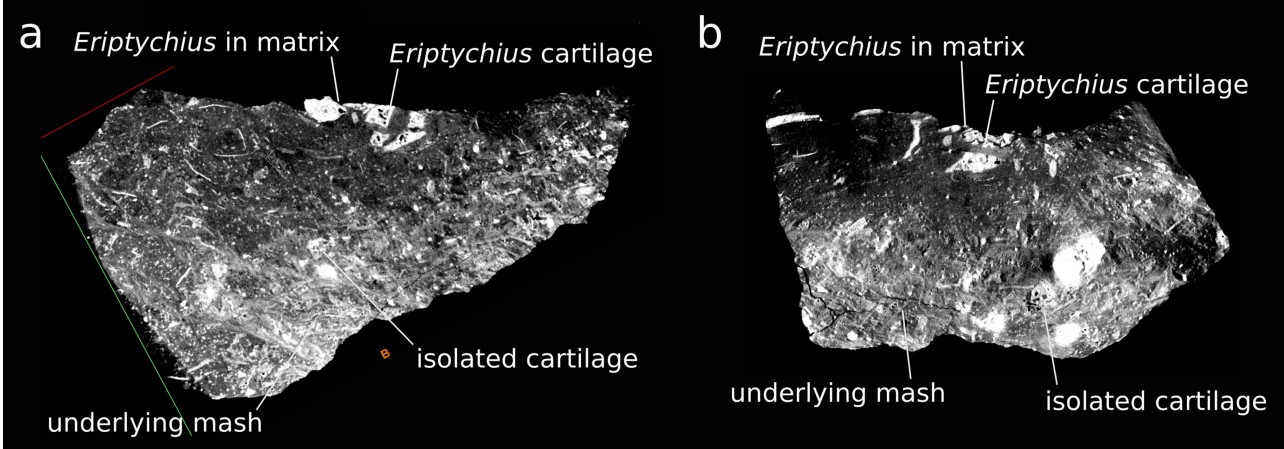

**Extended Data Fig. 1 | *Eriptychius* PF 1795 shown in the context of the surrounding matrix.** a,b, Tomograms showing sections through the part of the specimen preserved in the matrix. c–e, 3D render of the articulated *Eriptychius* specimen part preserved in epoxy relative to the matrix part, (c) in anatomical ventral view, (d) lateroventral view and (e) laterally with the matrix rendered transparent.

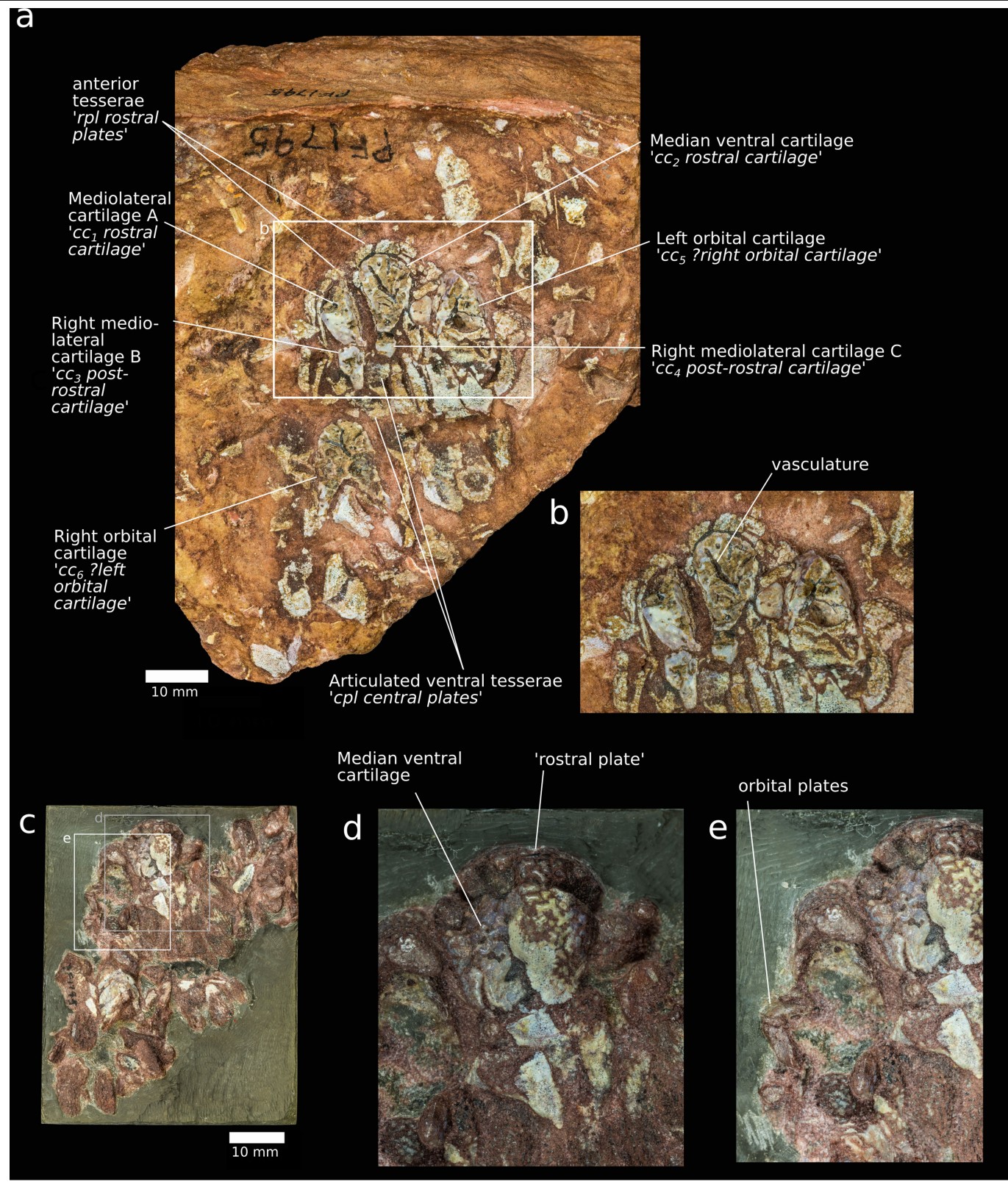

**Extended Data Fig. 2 | Photographs of *Eriptychius* PF 1795.** a, part preserved in matrix contrasting our interpretation with that of Denison[8]. b, a close up of the front part of the specimen showing the vasculature. c, the part of the specimen preserved in epoxy with d, a close up of the 'rostral plates and e, a close up of the orbital plates. Roman case labels, our interpretation; italics case labels, Denison's interpretation corresponding to figure 2 of ref.8.

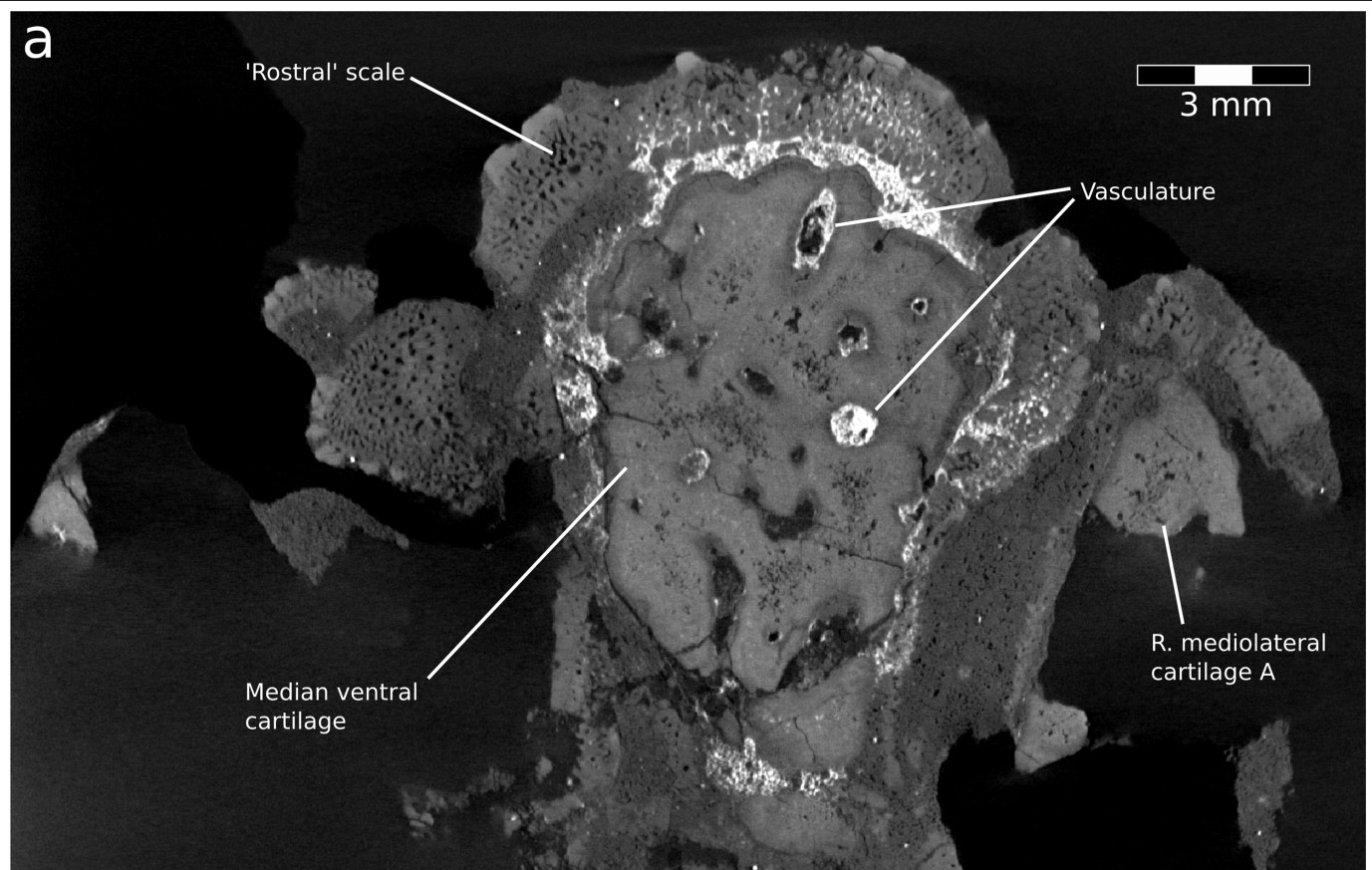

**Extended Data Fig. 3 | *Eriptychius* PF 1795 tissues in tomographic section.** Section of tomogram from the higher resolution scan set in coronal plane showing dermal 'rostral' plates overlying and wrapping cartilage.

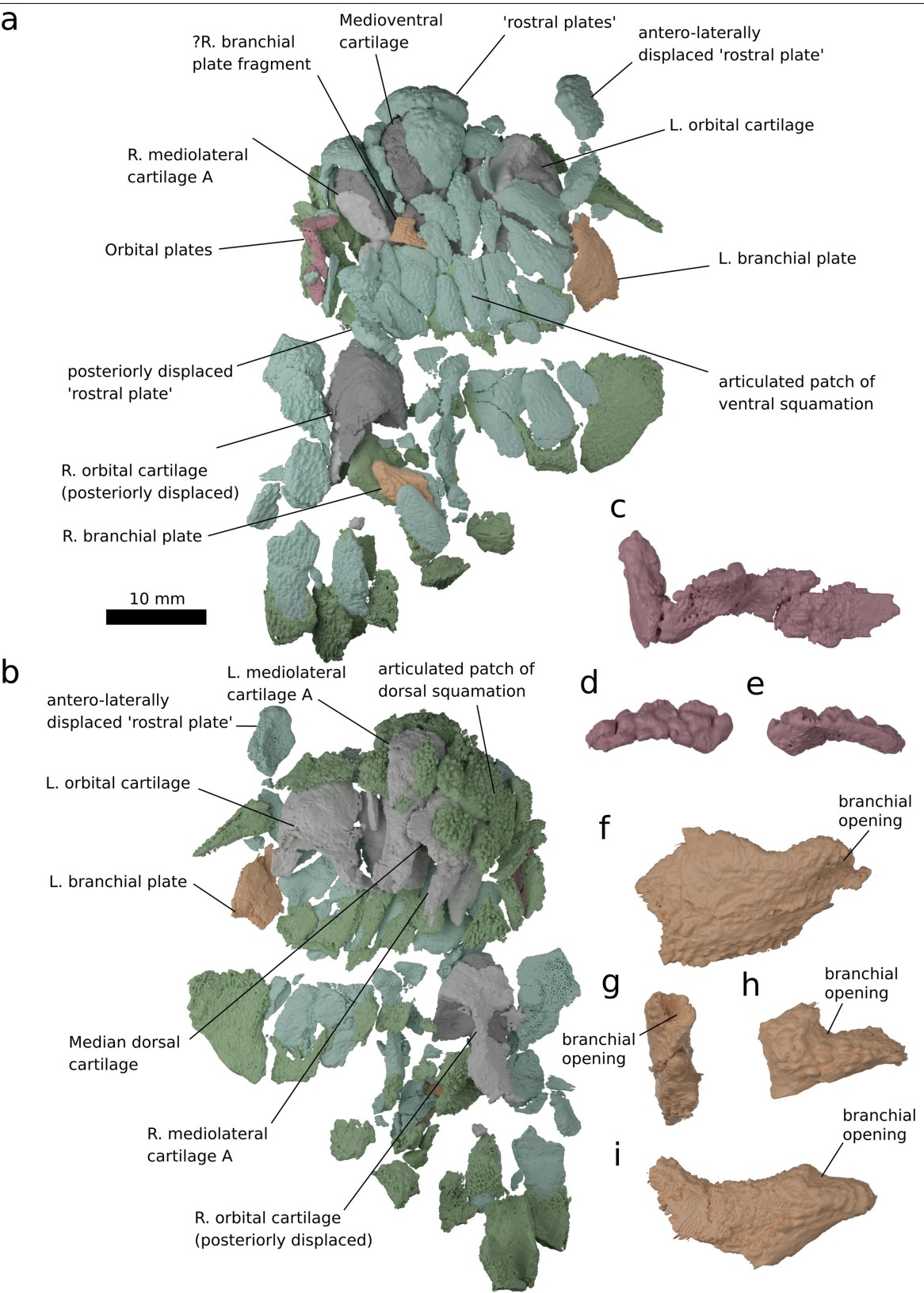

**Extended Data Fig. 4 | 3D render of *Eriptychius* PF 1795.** a, in ventral view. b, in dorsal view. c, chain of orbital plates in visceral view, showing curved visceral surfaces. d,e, the best-visualized orbital plate in (d) lateral and (e) visceral view. f, g, left branchial plate in (f) lateral and (g) anterior view. h, ?R. branchial plate fragment in lateral view. i, right branchial plate in lateral view. Colour scheme and abbreviations as in Figs. 1 and 2, except: grey, cartilages; blue-green, dermal skeleton. Lighter shades denote material from PF 1795a, darker shades PF 1795b.

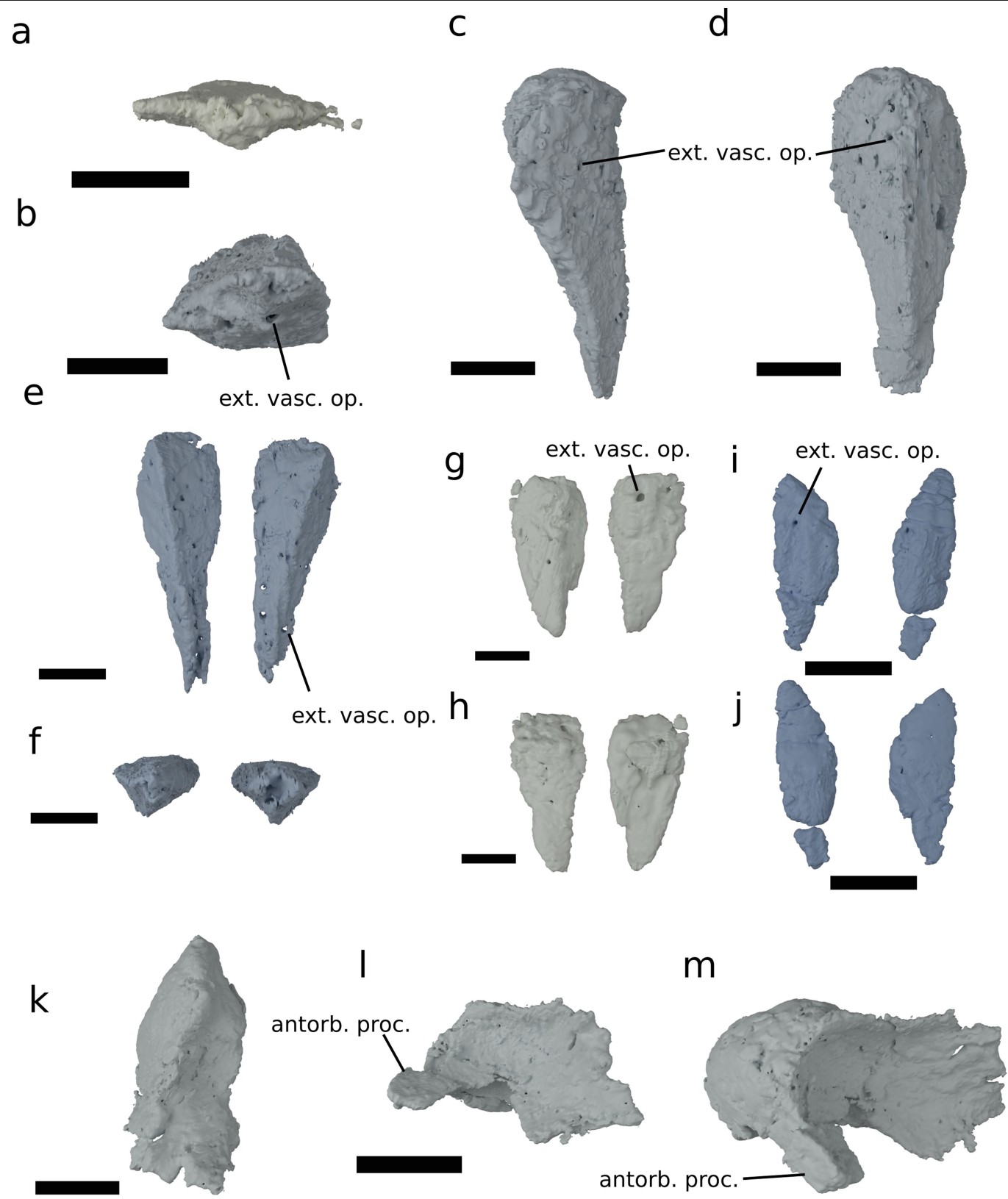

**Extended Data Fig. 5 | 3D renders of the individual endocranial elements of**
***Eriptychius* PF 1795.** a, median dorsal cartilage in anterior view; b-d median
ventral cartilage in (b) posterior, (c) left lateral and (d) dorsal view; e,f mediolateral
cartilages A in (e) ventral and (f), posterior view; g,h both surfaces of mediolateral
cartilages B; i,j both surfaces of mediolateral cartilages C; k-m, left orbital
cartilage in (k) medial, (l) posterior and (m) latero-posterior view. Colour
scheme and abbreviations as in Fig. 2.

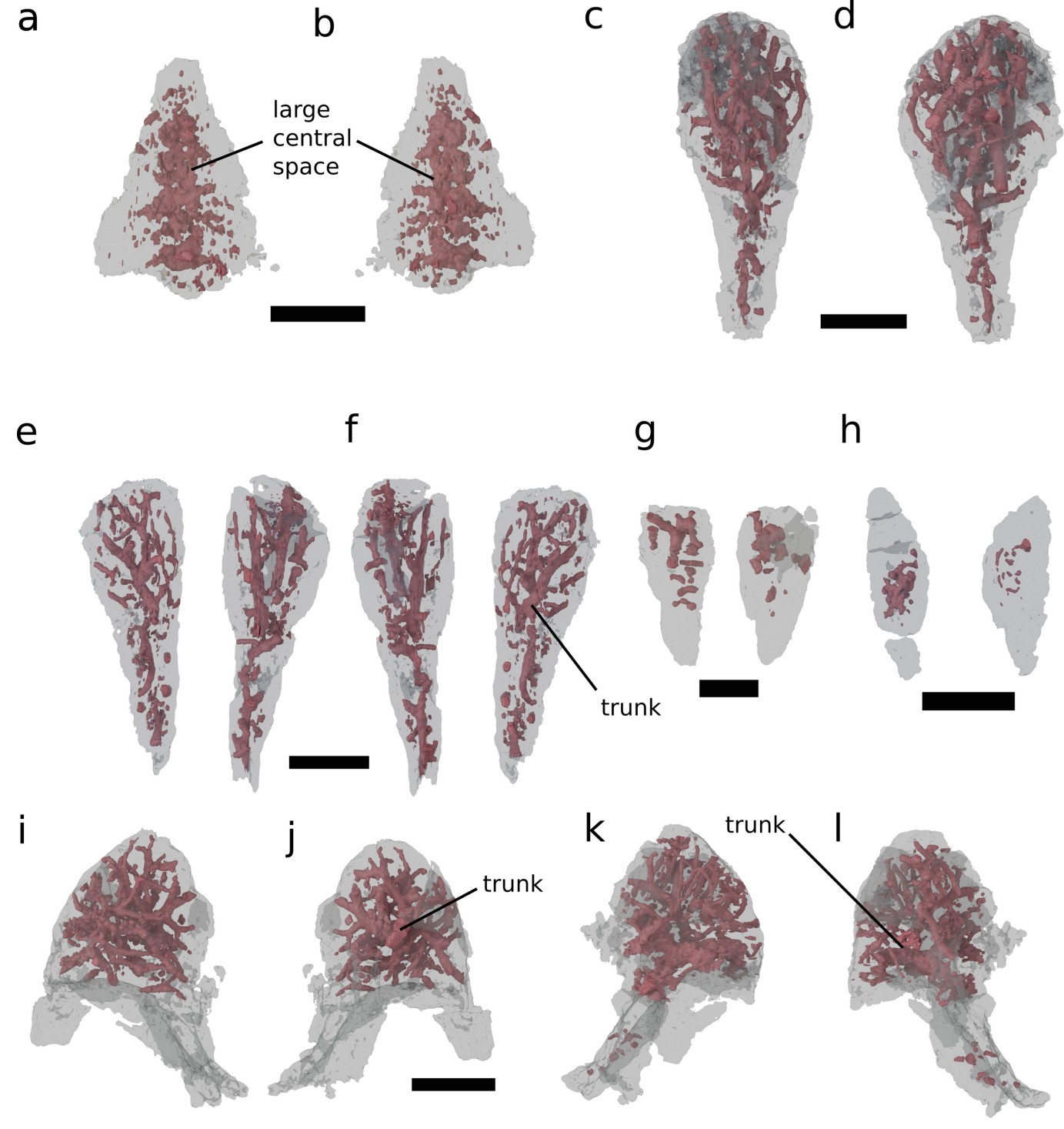

a

b

large
central
space

c

d

e

f

g

trunk

h

i

j

trunk

k

trunk

l

**Extended Data Fig. 6 | Additional 3D renders of the vascularization of the endocranium of *Eriptychius* PF 1795.** a,b, median dorsal cartilage in (a) dorsal and (b) ventral view; c,d, median ventral cartilage in (c) ventral and (d) dorsal view; e,f, mediolateral cartilages A in (e) dorsal and (f) ventral view; g, mediolateral cartilages B; h, mediolateral cartilages C; i,j left orbital cartilage in (i) dorsal and (j) ventral view; k,l right orbital cartilage in (k) dorsal and (l) ventral view. Colour scheme as in Fig. 2.

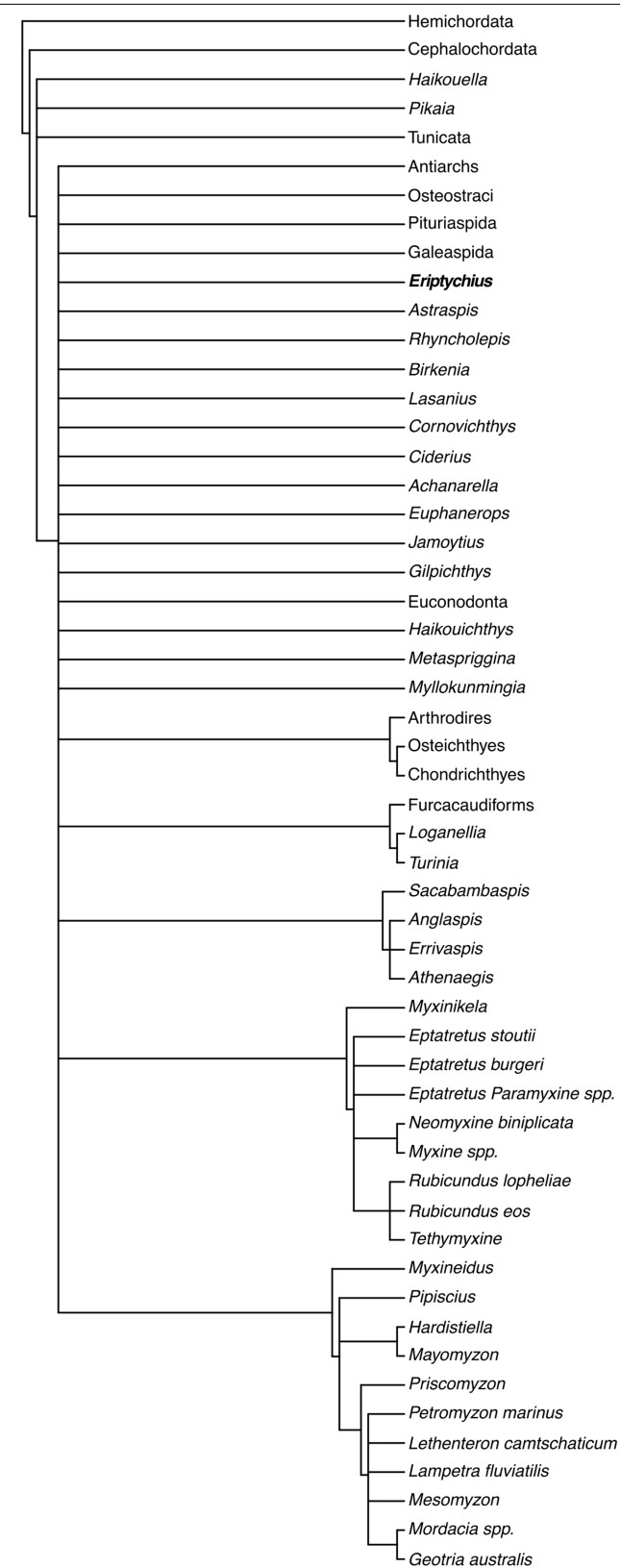

**Extended Data Fig. 7 | Strict consensus result of the parsimony analysis.**
Strict consensus tree resulting from the parsimony phylogenetic analysis
described in the methods.

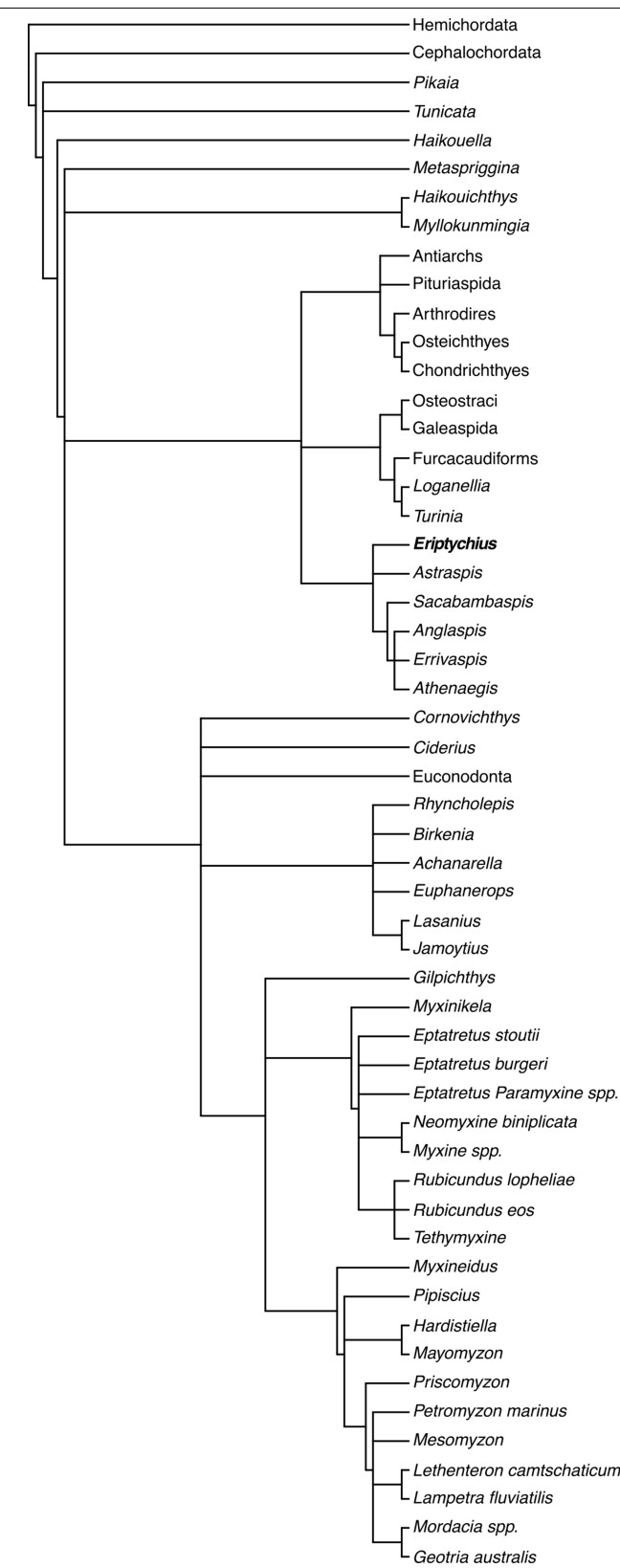

**Extended Data Fig. 8 | Adams consensus result of the parsimony analysis.**
Adams consensus tree resulting from the parsimony phylogenetic analysis
described in the methods.

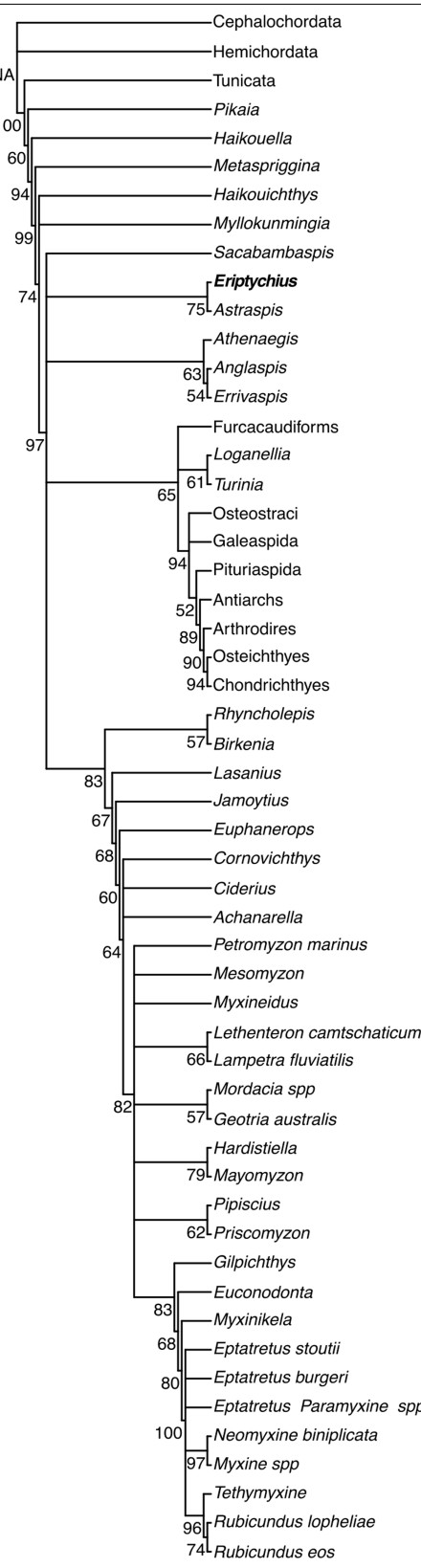

**Extended Data Fig. 9 | Consensus result of the Bayesian analysis.** Majority rule consensus tree resulting from the Bayesian phylogenetic analysis described in the methods. Node values correspond to posterior probabilities.

# Reporting Summary

## Statistics

For all statistical analyses, confirm that the following items are present in the figure legend, table legend, main text, or Methods section.

| n/a | Confirmed | |
|---|---|---|
| ☒ | ☐ | The exact sample size (*n*) for each experimental group/condition, given as a discrete number and unit of measurement |
| ☒ | ☐ | A statement on whether measurements were taken from distinct samples or whether the same sample was measured repeatedly |
| ☒ | ☐ | The statistical test(s) used AND whether they are one- or two-sided<br>*Only common tests should be described solely by name; describe more complex techniques in the Methods section.* |
| ☒ | ☐ | A description of all covariates tested |
| ☒ | ☐ | A description of any assumptions or corrections, such as tests of normality and adjustment for multiple comparisons |
| ☒ | ☐ | A full description of the statistical parameters including central tendency (e.g. means) or other basic estimates (e.g. regression coefficient) AND variation (e.g. standard deviation) or associated estimates of uncertainty (e.g. confidence intervals) |
| ☒ | ☐ | For null hypothesis testing, the test statistic (e.g. $F$, $t$, $r$) with confidence intervals, effect sizes, degrees of freedom and $P$ value noted<br>*Give P values as exact values whenever suitable.* |
| ☐ | ☒ | For Bayesian analysis, information on the choice of priors and Markov chain Monte Carlo settings |
| ☒ | ☐ | For hierarchical and complex designs, identification of the appropriate level for tests and full reporting of outcomes |
| ☒ | ☐ | Estimates of effect sizes (e.g. Cohen's *d*, Pearson's *r*), indicating how they were calculated |

*Our web collection on statistics for biologists contains articles on many of the points above.*

## Software and code

Policy information about availability of computer code

| Data collection | not applicable |
|---|---|
| Data analysis | For analysis of the image data we used the paid software Mimics v.25 (Materialise) to make 3D models and the free software Blender (blender.org) to acquire images. For the phylogenetic analysis we used the software tnt and MrBayes. |

For manuscripts utilizing custom algorithms or software that are central to the research but not yet described in published literature, software must be made available to editors and reviewers. We strongly encourage code deposition in a community repository (e.g. GitHub). See the Nature Portfolio guidelines for submitting code & software for further information.

## Data

Policy information about availability of data

All manuscripts must include a data availability statement. This statement should provide the following information, where applicable:

- Accession codes, unique identifiers, or web links for publicly available datasets
- A description of any restrictions on data availability
- For clinical datasets or third party data, please ensure that the statement adheres to our policy

The CT scan data this work is based on is available via Morphosource via the following links: PF 1795a (https://doi.org/10.17602/M2/M510644), PF 1795b (https://doi.org/10.17602/M2/M510665). A mesh incorporating all 3D surface models generated in this study is made available via Morphosource at the following link

(https://doi.org/10.17602/M2/M542071). Individual 3D meshes and the phylogenetic matrix are made available on figshare at the following links (https://doi.org/10.6084/m9.figshare.23726487).

# Research involving human participants, their data, or biological material

Policy information about studies with [human participants or human data](). See also policy information about [sex, gender (identity/presentation), and sexual orientation]() and [race, ethnicity and racism]().

| | |
|---|---|
| Reporting on sex and gender | not applicable |
| Reporting on race, ethnicity, or other socially relevant groupings | not applicable |
| Population characteristics | not applicable |
| Recruitment | not applicable |
| Ethics oversight | not applicable |

Note that full information on the approval of the study protocol must also be provided in the manuscript.

# Field-specific reporting

Please select the one below that is the best fit for your research. If you are not sure, read the appropriate sections before making your selection.

☐ Life sciences  ☐ Behavioural & social sciences  ☒ Ecological, evolutionary & environmental sciences

For a reference copy of the document with all sections, see [nature.com/documents/nr-reporting-summary-flat.pdf]()

# Ecological, evolutionary & environmental sciences study design

All studies must disclose on these points even when the disclosure is negative.

| | |
|---|---|
| Study description | We used computed tomography to image a fossil fish and then segmented the data to create 3D models of its skeleton. We used these as the basis for interpretation and phylogenetic analysis. |
| Research sample | Eriptychius americanus specimen from the Field Museum, Chicago PF 1795 |
| Sampling strategy | Only one specimen exists. |
| Data collection | The specimen was CT scanned at the UChicago PaleoCT facility by April Neander, and subsequently at Bristol University, UK by Sam Giles and Liz Martin-Silverstone |
| Timing and spatial scale | This study was carried out between June 2022 and March 2023 |
| Data exclusions | No data were excluded |
| Reproducibility | CT scan settings are given in the materials and methods, the scan data and phylogenetic data is provided freely allowing others to reproduce the study. |
| Randomization | This is not relevant as only one specimen was used |
| Blinding | Blinding is irrelevant as no blinding was possible |

Did the study involve field work?  ☐ Yes  ☒ No

# Reporting for specific materials, systems and methods

We require information from authors about some types of materials, experimental systems and methods used in many studies. Here, indicate whether each material, system or method listed is relevant to your study. If you are not sure if a list item applies to your research, read the appropriate section before selecting a response.

## Materials & experimental systems

| n/a | Involved in the study |
|-----|----------------------|
| ☒ | Antibodies |
| ☒ | Eukaryotic cell lines |
| ☐ ☒ | Palaeontology and archaeology |
| ☒ | Animals and other organisms |
| ☒ | Clinical data |
| ☒ | Dual use research of concern |
| ☒ | Plants |

## Methods

| n/a | Involved in the study |
|-----|----------------------|
| ☒ | ChIP-seq |
| ☒ | Flow cytometry |
| ☒ | MRI-based neuroimaging |

## Palaeontology and Archaeology

Specimen provenance — The specimen is held in the Field Museum, Chicago, USA and was taken out on loan for the duration of the study.

Specimen deposition — The specimen continues to be held at the Field Museum

Dating methods — No new dates are provided

☐ Tick this box to confirm that the raw and calibrated dates are available in the paper or in Supplementary Information.

Ethics oversight — None required, material already held in collections

Note that full information on the approval of the study protocol must also be provided in the manuscript.

