## [Peer Review File · Nature]

Manuscript Title: The oldest three-dimensionally preserved vertebrate neurocranium

Reviewer Comments & Author Rebuttals

Reviewer Reports on the Initial Version:

Referees' comments:

Referee #1 (Remarks to the Author):

Review of "The oldest three-dimensionally preserved vertebrate cranial endoskeleton" by Dearden et al.

The vertebrate cranial endoskeletons are rich sources of morphological data, which have been used repeatedly to build hypotheses of relationships of Early vertebrates. However, the cranial endoskeletons of early vertebrates remain poorly known because they are made entirely of cartilage and only can be exceptionally preserved in rare condition. Using computed tomography (CT) scanning, Dearden et al. describe the oldest three-dimensionally preserved vertebrate cranial endoskeleton in an Ordovician stem-group gnathostome Eriptychius from the Harding Sandstone of Colorado, USA. They revealed the preorbital part of the head with a set of paired and midline cartilages in the oldest gnathostome group for the first time. It is very interesting that the authors have identified four paired cartilages symmetrical across the midline and the median ventral cartilage in the preorbital part of the head of Eriptychius. This is an extremely important discovery with broad implications for our understanding the origin of gnathostome trabeculae cranii which has long been regarded as a major developmental and evolutionary advance over agnathans, and certainly appropriate for publication in Nature. However, the authors don't seem to be fully aware of the potential significance for their so important findings. They think these structures are distinct from that of all known cyclostomes and gnathostome, and probably represent a third way of constructing the vertebrate cranial endoskeleton. From my view, these structures may exactly compare to the derivative structures of trabeculae in galeaspid and gnathostomes. Therefore, they should bear a much more important evolutionary significance for our understanding the origin of gnathostome head. I strongly recommend this paper for publication after a major revision. Some general comments:

Line 15: gnathostomes = jawless 'ostracoderms' (stem gnathostomes) +jawed vertebrates. Maybe, crown gnathostomes is better

Line 26: coalescence of cartilages into a single neurocranial unit is a derived trait

Here, maybe the 'skull' is better than 'neurocranial unit' because the coalescence of cartilages into a skull comprising the neurocranial cartilages and splanchnocranial cartilages.

Line 37-38 In living cyclostomes the cranial endoskeleton comprises an open trabecular structure holding the brain, and a feeding apparatus formed of a symmetrical set of paired and midline cartilages.

The trabecular structure in lampreys surrounds the hypophysis and extends anteriorly to the olfactory capsules. They are lateral rather than anterior to the notochord, and probably represent the anterior prolongation of parachordal cartilage. The developmental data indicate that the 'lamprey trabecula' develops from the mandibular mesoderm, and is not homologous with the

gnathostome trabecula, which develops from premandibular neural crest cells.

Line 39-40 brain instead enclosed by a neurocranium, a single endoskeletal structure that encases the brain and nasal capsules, with paired mandibular and hyoid arches forming the feeding apparatus.

Yes, it is a major innovation that the trabeculae cranii and the visceral skeleton (mandibular and hyoid arches) are incorporated into cranial endoskeleton in gnathostomes.

Line 46-47 the plesiomorphic state for gnathostomes is a single, jawless neurocranial unit filling the space between the brain and dermal skeleton enclosing the brain and pharynx.

In galeaspids and osteostracans, the lateral visceral skeleton (splanchnocranium) merged with the middle braincase (neurocranium) to form a massive endoskeletal skull enclosing the brain and pharynx. Therefore, the endoskeleton filling the space between the brain and dermal skeleton includes the neurocranial unit+splanchnocranial unit. Maybe, 'endoskeletal unit' or 'skull' is better than 'neurocranial unit'.

Line 48-50 In the multiple other groups of Palaeozoic vertebrates all that is known of the cranial endoskeleton comes from difficult-to-interpret two-dimensional fossils or is inferred based on the dermal skeleton.

Yes, there are no evidence for the cranial endoskeleton in heterostracans. Janvier (1996) thought that the endoskeleton, whether cartilaginous or calcified, is probably absent in heterostracans because the internal surface of the armor plates is generally quite smooth and shiny, with a few nutrient canals, even at places marked with internal impressions of organs. In addition, the fact that the brain and labyrinth could leave such an accurate impression on the dermal skeleton suggests that they were enclosed only in fibrous sheath probably without intervening cartilage.

Line 54: Sandbian (458.4-454.0 Ma) should be (458.4-453.0 Ma)

Line 60-62: The cranial cartilages of Eriptychius are anatomically dissimilar from the crania of both cyclostomes and gnathostomes, revealing the existence of a third way of skeletonising the vertebrate cranium.

Maybe, it does not reveal the existence of a third way of skeletonising the vertebrate cranium, but represents a very, very important intermediate state before galeaspid and osteostracans, in which the gnathostome independent trabeculae cranii has formed, but the visceral skeleton (splanchnocranium) has not fused with the neurocranium to form a single massive endoskeletal skull. The fusion of the derivative structures of trabeculae and the visceral skeleton (splanchnocranium) with braincase happened later in galeaspids and osteostracans. See the comments for Line 125-150 for more details.

Line 73: Eriptychius americanus belongs to the stem-gnathostomata, not Agnatha any more

Line 74,93.....: The head dermal bone maybe use 'plates' better than 'scales'

Line 125-150: About the interpretation for the paired orbital cartilages, three paired cartilages symmetrical across the midline – two dorsal (termed mediolateral A, B) and one ventral (termed mediolateral C), and the median ventral cartilage.

It is very interesting that the authors have identified four paired cartilages symmetrical across the midline and the median ventral cartilage, which are distinct from that of known cyclostomes and gnathostomes, and probably represent a third way of constructing the vertebrate cranial endoskeleton. From my view, these structures may exactly compare to the derivative structures of trabeculae in galeaspids and gnathostomes. In crown gnathostomes, the trabeculae derivative structures (e.g. orbitonasal lamina, postnasal wall, ethmoid process, interorbital septum, suborbital shelf, supraorbital crest, and rostrum) are fused with the anterior part of neurocranium (chordal

cranium) to form the trabeculae cranii (prechordal cranium). In crown gnathostomes, the trabeculae cranii and derivative structures are of neural crest origin, developed mainly from the premandibular ectomesenchyme (Kuratani et al. 2001). However, the corresponding premandibular crest cells in lampreys developed into the upper lip and the floor of the nostril, or the nasohypophysial duct (Kuratani et al. 2001; Kuratani 2004). Therefore, they should bear a much more important evolutionary significance for our understanding the origin of gnathostome trabeculae cranii.

1) the paired orbital cartilages may compare to the cartilages of supraorbital crest in galeaspids and chondrichthyans.

In chondrichthyans, the supraorbital crest is an arched horizontal plate that extends posteriorly along the dorsal edge of the medial orbital wall from the preorbital process to the postorbital process, with its base continuous with the orbital wall and cranial roof (Compagno 1999). The supraorbital crest is penetrated medially by a row of supraorbital foramina for the tiny branches of the superficial ophthalmic nerve supplying the supraorbital lateral line canal (de Beer 1937). The supraorbital crest and the dorsal edge of the interorbital wall are delimited by the row of supraorbital foramina (de Beer 1937). Like crown gnathostomes, the supraorbital crest with a row of supraorbital foramina has been clearly identified in galeaspids. By contrast, the supraorbital crest is lacking in hagfishes and lampreys. The embryonic development of crown gnathostomes shows that the supraorbital crest is formed in the last stage (stage 6, 90mm, the skull has taken on the appearance of that of the adult) (de Beer 1937).

2) the paired mediolateral cartilages A may compared to the cartilages of the medial orbital wall (the interorbital septum) in galeaspids and chondrichthyans.

The medial wall of orbital cavity or the interorbital septum has been identified in galeaspids and chondrichthyans. In both groups, the medial orbital wall is delimited from the supraorbital crest by the row of supraorbital foramina. They are penetrated by numerous canals for the cranial nerves and vascular canals which show a similar arrangement. As the derivatives of trabeculae cranii (innovation of gnathostomes), the medial orbital walls are absent in hagfishes and lampreys.

3) the mediolateral cartilages C may compare to the cartilages of the suborbital shelf in galeaspids and chondrichthyans

In most chondrichthyans, the orbital cavity is bordered ventrally by the suborbital shelf, which tapers abruptly toward the midline to join its counterpart in a blunt point below the rostrum (Schaeffer 1981; Compagno 1999). The orbital cavity is also lined ventrally by a broad suborbital shelf in placoderms (Goujet 1984a). By contrast, the braincase of osteichthyans and fossil acanthodians is rather narrow and deep, with ventrally open orbital cavities without a suborbital shelf (Janvier 1996a). Like chondrichthyans and placoderms, the orbital cavity of galeaspids is lined ventrally by a broad cartilaginous lamina for the suborbital shelf which separated the orbital cavity from the oralbranchial chamber completely.

4) the mediolateral cartilages B in Eriptychius may compare to the cartilages of orbitonasal lamina in galeaspids and chondrichthyans (not sure) .

In chondrichthyans, the trabeculae cranii grow forward into the rostral region to form the orbitonasal lamina, which surrounds the nasal sacs on each side, as well as the rostral cartilages (Jollie 1971). This structure lies between the anterior eye margin and nasal sac and forms the hind wall and part of the side wall of the nasal capsule (de Beer and Moy-Thomas 1935; de Beer 1937). The position of the orbitonasal lamina is also marked by the foramina for the profundus nerve (V1.f, Fig. 5.7A) and the so-called orbitonasal canal through which the orbitonasal vein (anterior facial vein) leaves the orbital cavity rostrally before entering the nasal capsule. de Beer (1937) pointed out

that the existence of the ectethmoid chamber and of the orbitonasal foramen is evidence of the fact that the orbitonasal lamina is not an original part of the side wall of the cranial cavity, but is the original posterolateral wall of the nasal capsule, which has secondarily contacted with the wall of the cranial cavity. The orbitonasal lamina has been identified in galeaspids because the nasal capsules are located lateral to the base of the rostrum and rostral to the orbital cavity. Furthermore, the foramina for the profundus nerve, the so-called orbitonasal canal, and the canal for orbitonasal artery have also been identified in galeaspids. These foramina and canals in the orbitonasal lamina are almost the same as seen in gnathostomes. Thus, coupled with the position of the nasal capsules, the orbitonasal lamina is present in galeaspids. As the derivatives of trabeculae cranii (an innovation of gnathostomes), the orbitonasal lamina has never been identified in other jawless vertebrates because the nasal capsule of hagfishes, lampreys, and osteostracans is located high on the dorsal surface of the head, it is impossible to form a lamina between the orbital cavity and nasal capsules. Therefore, the orbitonasal lamina is lacking in hagfishes, lampreys, and osteostracans. In Eriptychius, the paired nasal capsules is probably extremely terminal and located in a space between the paired mediolateral cartilages A and the median ventral cartilage. The paired mediolateral cartilages B in Eriptychius is small and short, and maybe can be regarded as the precursor of the orbitonasal lamina.

5) The median ventral cartilage may compare to the cartilage of ethmoid plate (process) in galeaspids and chondrichthyans

In galeaspids, the orbitonasal lamina protrudes forward as a median cartilaginous rod separating and partly floored the paired olfactory bulbs. As the median cartilaginous rod is positioned just rostral to the hypophysial opening and ventral to forebrain, it is most readily compared to the ethmoid plate of chondrichthyans. In chondrichthyans, the paired trabeculae fuse with one another ventral to the brain, and project forward to form the rostral carina or ethmoid plate supporting the forebrain and rostrum ventrally (de Beer 1937; Wake 1979; De Iuliis and Pulerà 2007). The ethmoid plate (et.p, Fig. 5.7F) further forms a vertical internasal septum between the nasal capsules (Wake 1979). However, compared to chondrichthyans, the ethmoid plate of galeaspids only supported the forebrain ventrally, but does not extend forward enough to form the internasal septum separating the paired nasal capsules. Judging from the position of olfactory bulb, the median ventral cartilage in Eriptychius is most likely corresponding to the ethmoid plate (process) in galeaspids and chondrichthyans.

Line 131-133 : The anteriormost branchial scale lies postero-lateral to the other orbital cartilage (Figs 1, S3), putting the orbits and the first branchial arch in the same relative positions as other jawless stem-gnathostomes and cyclostomes.

This indicates that the branchial arches of Eriptychius are independent, and are not incorporated into the neurocranium to form a massive skull as in osteostracans and galeaspids.

Line 152-155: All cartilages are pervasively penetrated by canals (Fig. 2g, S6). In the larger cartilages, i.e. the orbital cartilages and the mediolateral cartilages A, this tends to follow the pattern of a larger trunk entering the cartilage from the posterior side before splitting into smaller branches that open to the surface.

Maybe, these canals can be interpreted as the tiny branches of the superficial ophthalmic nerve supplying the supraorbital lateral line canal (de Beer 1937), which supporting the interpretation of the orbital cartilages and the mediolateral cartilages A as supraorbital crest and the medial orbital wall respectively.

Line 203-207 The cranial cartilages of Eriptychius are not obvious homologues of endoskeletal

structures in the head of any known extant or extinct vertebrate (Figs 2, S4).

The cranial cartilages of *Eriptychius* are probably homologues of the derivative structures of gnathostome trabeculae cranii. This indicates that the gnathostome trabeculae cranii arose as early as the Sandbian (458.4-454.0 Ma). This corroborated that the derivative structures of gnathostome trabeculae cranii are primitively independent as indicated by the development of gnathostomes. The coalescence of cranial cartilages and visceral arches into a single skull is a derived trait in galeaspid, and osteostracans, and jawed vertebrates.

Line 220-222 This essentially appears to be the role of the cartilages in *Eriptychius*, although, unlike osteostracans and galeaspids, there is no evidence for any mineralisation from the level of the orbits posteriorly.

In galeaspids, the dorsal portion of branchial arches are incorporated into the neurocranium to form a massive skull (Gai et al, 2022) as assumed in osteostracans by Stensiö (1927, 1964). It has been known that the mandibular arch is located directly behind the orbital cavity and formed the posterior wall of orbital cavity. According to the authors, the anteriormost branchial scale lies postero-lateral to the other orbital cartilage (Figs 1, S3), putting the orbits and the first branchial arch (mandibular arch) in the same relative positions as other jawless stem-gnathostomes. This indicates that the branchial arches of *Eriptychius* are independent, and are not incorporated into the neurocranium to form a massive skull as in osteostracans and galeaspids. This may explain why the orbits of *Eriptychius* remain unclosed posteriorly and no evidence for any mineralisation from the level of the orbits posteriorly.

Line 269 Figure 3C, Supplementary information Character 12, Character 38

A paired 'pineal' opening has been reported in the arandaspids *Arandaspis* (Ritchie & Gilbert-Tomlinson 1977) and *Sacabambaspis* (Gagnier 1993). However, the paired openings lie well behind the eyes, which are an unusual position for such structures. In addition, the paired openings are arranged in bilateral symmetry, which is very similar the condition of the paired endolymphatic openings in galeaspids, osteostracans, placoderms, and chondrichthyans on the dorsal surface of the head. So it is more likely that the paired 'pineal' openings in *Sacabambaspis* and *Arandaspis* are the paired endolymphatic openings. The real pineal opening of *Arandaspis* is a single opening and probably located in the single median pineal plate between the two eyes since the orbits are placed at the extreme anterior margin of the headshield in *Arandaspis* and *Sacabambaspis*. In lampreys, arandaspids, some heterostracans, anaspids, osteostracans, galeaspids, and placoderms, the pineal opening is a single structure perforating the dermal skull roof.

Referee #2 (Remarks to the Author):

This manuscript provides the first 3D description of a long enigmatic skull of the Ordovician vertebrate *Eriptychius*, from the famous Harding Sandstone Formation of Colorado (USA), which yields some of the earliest vertebrates known to date. This specimen, first recorded by Denison in 1967, is an assemblage of elements of calcified cartilage pervaded by blood vascular canals, and odontode-bearing dermal bones, which was extremely difficult to interpret. It is now possible thanks

to CT scanning which allows to make sense out of this crushed mix of bones and cartilage. Eriptychius, like most other vertebrates from the Harding Sandstone, was classically regarded as an “ostracoderm”, that is, a jawless stem gnathostome, and gathered with heterostracans in the pteraspidomorphs, based on the reputedly-yet debated- acellular structure of the dermal bone. This detail description of the first vertebrate endoskeleton, although somewhat puzzling for those who keep in mind the braincase structure of e.g. Silurian-Devonian or extant vertebrates, provides an unprecedented source of documentation for reconstructing vertebrate phylogeny, which the authors have attempted here masterly, and with insightful views on character transformations in early vertebrate history.

Referee #3 (Remarks to the Author):

The moment I saw the figures I immediately grasped the importance of this paper. That the specimen was spotted (by Denison) and reconstructed to this detail (by the authors) is just as stunning as that it exists. I myself pulled hundreds of drawers containing blocks of Harding Sandstone sclerites, nonchalantly – and almost jokingly – scanning the surfaces for a surprise find. “Another Eriptychius graveyard,” I’d tell myself and move on after a few seconds of optical scanning. In fact, I’ve done that just a few days before receiving this manuscript. Now I’m sure that somewhere there I must have missed specimens like this under my nose. Nicely scored!

I sustain the major points from the authors as well justified. These are:

- The earliest three-dimensionally preserved vertebrate endocranium
- Independent cartilages forming the anterior part of that endocranium, distinct from major patterns known in other vertebrate lineages
- Dimensions and configurations of these skeletal parts support, and to some extent reflect, primitive patterns of the vertebrate skull as implied by recent phylogenetic shakeups around the vertebrate crown node.

To enhance this paper, I offer several major suggestions for the authors to consider, and a long list of minor comments that follows.

1) The authors have an unparalleled opportunity here to address important predictions of the New Head Hypothesis (DOI: 10.1126/science.220.4594.268) and the Heterotopy Hypothesis (DOI: 10.1111/j.1525-142X.2011.00523.x), the former in the context of vertebrate origins and the latter in gnathostome evolution.

The New Head Hypothesis predicts that the prechordal cranium (an extensive forebrain and its skeletal capsule) is a vertebrate novelty with the origin of neural crest. Here, as the authors point out, the prechordal cranium occupies minimal dimensions in this stem gnathostome as in Cambrian stem vertebrates and Devonian stem cyclostomes (euphaneropids), suggesting that: a) the “new head” was diminutive at best at the origin of the vertebrate crown; and b) the extensive prechordal crania evolved independently in cyclostomes and gnathostomes. A diminutive prechordal cranium in Cambrian stem vertebrates has been considered preliminarily in the context of neural crest origins (DOI: DOI: 10.1126/science.adc9198; DOI: 10.1017/9781316832172.003).

The Heterotopy Hypothesis predicts that the origins of paired nasal capsules and trabecula cranii were linked, and were each a necessary condition for the origin of jaw skeleton. But here, *Eryptichius* appears to have paired nasal capsules (like *Haikouichthys*) and no discernible trabecula cranii or jaws. Might this imply that a) paired nasal capsules are the ancestral for the vertebrate crown; and b) the monorhiny condition independently evolved in cyclostomes and jawless stem gnathostomes (therefore cyclostomes, just like the case of filter-feeding larval lampreys, fail once again to serve as a surrogate ancestor with respect to gnathostomes)?

2) I believe the authors can do more to emphasize their closing (reducing) the temporal gap of the vertebrate braincase record. *Haikouichthys* (DOI: 10.1038/nature01264) – taken at the face value – represents the earliest ‘vertebrate’ with its neurocranium preserved. The next one up has so far been the galeaspid *Shuyu* from the Middle Silurian. So there has been nearly a 100-my gap of this information across when all major vertebrate lineages arose up to osteichthyan crown. It’s amazing to have any endocranial information with one of the earliest vertebrates with mineralized skeletons.

3) Although it doesn’t take much more to convince me of their anatomical reconstruction, I believe it would benefit the paper to strengthen their argument that the mineralized chondrocranial elements of *Eryptichius* truly forms the front part of the braincase. With the position of the proposed pineal foramen, and with the reconstruction of the olfactory bulbs and eyes, I have some difficult time wrapping my head around. I understand the challenging nature of this, but can the authors find additional evidence for the orbital identity (e.g., dermal scales preserved around the “orbital cartilages”; elaborating possible muscular attachment on orbital cartilage)? Any inference for nasal openings (e.g., penetration or demarcating on the paired fossae for “olfactory bulbs; break in dermal scales)? Notochordal sheath? I’m comfortable with the terminal mouth, and the authors made a good argument for this. Still, my first thoughts for the medial ventral cartilage were along the lines of “What if this is something like a hagfish’s subnasal cartilage, only pressed down to the ventral shield during fossilization?”

Because the endo- and exoskeletons are loosely connected, it seems to me that we can only be sure of their dorsoventral positions relative to each other, but not in absolute sense between the dorsal and ventral part of its head. There are free paired cartilages in both hagfish and lamprey heads in this region. Granted, these cartilages in *Eryptichius* look like none of those, and I’m not suggesting they correspond. What I would love to see here instead is evidence for the interaction between these cartilages and the nervous system. Much to ask, yes, but this would really strengthen their argument.

4) As an addendum to my points above, I have concerns over the terminologies such as “cranial endoskeleton” and “trabecular”. As for the former, I appreciate that the authors are trying to navigate carefully around the kind of problem I’m just about to point out on “trabecular”. However, their internal definition is unclear and I wonder if it's used softly in one place and rigidly in another. They should either: a) define at the outset what they mean by “cranial endoskeleton” and stick to that definition; or b) use more conventional, clearly defined terms such as chondrocranial elements, prechordal cranium, neurocranium, etc. As for “trabecular”, in a chondrocranium this element derives from the ectomesenchyme in a prechordal position. Therefore, as the authors acknowledge,

“trabecula” in cyclostomes are parachordal elements. Controlling this terminology will of course be important to address their findings in the context of the New Head and Heterotopy hypotheses.

5) I think it would be important to clearly illustrate discrepancies with respect to Denison’s observation. This may be done in a supplementary figure, where Denison’s reconstruction is presented side by side with the new scheme.

MINOR COMMENTS

Line 14: “... its form in the two living vertebrate groups...”

“... its form between the two living vertebrate groups...”

Lines 16-18: “the earliest three-dimensionally preserved vertebrate cranial endoskeletons are from Silurian-Devonian taxa that do little to resolve modern states.”

Here, I think it’s better to focus on the numbers. E.g.: Previously, no three-dimensionally preserved neurocranium has been known from the first 100 my of vertebrate evolutionary history.

Line 19: “...endoskeletal cranial anatomy...”

This is somehow awkward. Can it not be endocranial anatomy?

Line 25: “...more akin to crownward taxa...”

The authors mean “crownward gnathostomes” here, correct?

Line 30: “...existed in the earliest skeletonising jawed vertebrates.”

I’m not sure if this is correct. Perhaps “existed among the earliest known stem gnathostomes”?

Line 35: I’m not sure if “frustrated” is the word here. Perhaps “challenged”?

Line 37 and elsewhere: Watch out for the homology of trabecula cranii whenever this structure in cyclostome is discussed. The authors understand this, but not all readers do.

Line 38: THE feeding apparatus CONSISTING of a...

Lines 40-41: “...a single endoskeletal...capsules...”

Put this in parentheses.

Lines 42-44: “...to polarise these conditions the fossil record of stem-group gnathostomes has proved key to reconstructing their evolution, but does little to bridge the morphological gap between cyclostomes and gnathostomes.”

This needs to be more precise (and less repetitive). “...to polarise these conditions, the fossil record of stem-group gnathostomes would be critical to reconstructing their evolution, but has yielded little to bridge the morphological gap.”

Lines 45-46: “...the Silurian and Devonian placoderms, osteostracans, and galeaspids stem

gnathostomes together...”

“...the Silurian and Devonian placoderms, osteostracans, and galeaspids together...”

“...the Silurian and Devonian stem gnathostomes together...”

“...the Silurian and Devonian stem gnathostomes (placoderms, osteostracans, and galeaspids) together...”

Lines 47-48: Does a neurocranium “enclose” the pharynx? That would be splanchnocranium.

Lines 73-76: Emended diagnosis.

This first part of diagnosis is based on what would have been present in other Ordovician astraspids and eryptychiids but just happens to be preserved in this one taxon, so I’m not sure if this is appropriate for diagnosis of Eriptychius itself. This should refer to the histology of these elements (described in detail in Supplementary Text) and specific patterns discerned in the dermal skeleton of this taxon (which are mentioned toward the end of differential diagnosis).

Line 118 “further” should be “farther”

Lines 130-132: The relationship between the orbit and the first branchial arch varies among cyclostomes and stem gnathostomes (and has been described also in relation to the otic capsule, though that can only be distantly inferred here), so this needs some elaboration. How posterolateral?

Lines 152-166: Good addition and restrained comparison.

Line 203-214: I think this is quite fair.

Line 226: Please be clear about the “trabecular region” – do the authors mean a prechordal cranium here? Or something else?

Line 238: Cyclostomes also have a single neurocranium, only that it forms from the parachordal component (paraxial mesoderm) and has little contribution from the prechordal head.

Lines 243-254: I understand that the authors are navigating carefully to drive two complex points here: 1) Eriptychius offers the earliest evidence of a preorbital skeleton (=prechordal cranium?); but 2) the inferred anterior position of the orbits implies that a small or diminutive preorbital head is a widespread and potentially primitive condition among early vertebrates. I think these two points can be more explicitly or clearly delineated by driving home the position of the orbit as a landmark to delimit the preorbital/prechordal region.

Figure 3: panels a-e should be linked to the corresponding terminal taxa on the tree.

A generalised cyathaspid (d) is somewhat misleading because the grey shades (imprints on the dermal shield) cannot be distinguished from those for endoskeletal elements in the other taxa. Similarly, the dermal skull is not clearly distinguished against the endocranial elements in the osteostracan and placoderm (f, g) – why this is unshaded in Samabambapis, cyathaspid, and Eriptychius but shaded in Benneviaspis and Dicksonosteus?

There should be some labels to distinguish nasal apertures (b, c, f, g) against capsules (a, d, e?) because they occupy different positions.

Pineal (yellow) – colour code is difficult to make out.

Author Rebuttals to Initial Comments:

Referees' comments:

Referee #1 (Remarks to the Author):

Review of "The oldest three-dimensionally preserved vertebrate cranial endoskeleton" by Dearden et al.

The vertebrate cranial endoskeletons are rich sources of morphological data, which have been used repeatedly to build hypotheses of relationships of Early vertebrates. However, the cranial endoskeletons of early vertebrates remain poorly known because they are made entirely of cartilage and only can be exceptionally preserved in rare condition. Using computed tomography (CT) scanning, Dearden et al. describe the oldest three-dimensionally preserved vertebrate cranial endoskeleton in an Ordovician stem-group gnathostome Eriptychius from the Harding Sandstone of Colorado, USA. They revealed the preorbital part of the head with a set of paired and midline cartilages in the oldest gnathostome group for the first time. It is very interesting that the authors have identified four paired cartilages symmetrical across the midline and the median ventral cartilage in the preorbital part of the head of Eriptychius. This is an extremely important discovery with broad implications for our understanding the origin of gnathostome trabeculae cranii which has long been regarded as a major developmental and evolutionary advance over agnathans, and certainly appropriate for publication in Nature. However, the authors don't seem to be fully aware of the potential significance for their so important findings. They think these structures are distinct from that of all known cyclostomes and gnathostome, and probably represent a third way of constructing the vertebrate cranial endoskeleton. From my view, these structures may exactly compare to the derivative structures of trabeculae in galeaspids and gnathostomes. Therefore, they should bear a much more important evolutionary significance for our understanding the origin of gnathostome head. I strongly recommend this paper for publication after a major revision.

Many thanks to Referee #1 for their comments. We are pleased that they found our manuscript of interest and we agree with them that the new information that we provide is a unique new source of information for those building hypotheses about the origins of the gnathostome head. However, because the morphology and arrangement of these cartilages is unusual (and the fossil is incomplete), and the 480 million year gap between it and the present we are reluctant to impose a rigid identity on the cartilages based on the development of modern groups. Instead we want to construct a prudent hypothesis that will be testable by future researchers. We believe that we are justified in doing this as Referees #2 and #3 agree with the overall interpretation of the cartilages, with Referee #3 specifically describing the statement that they are unlike those of any other known

vertebrate as “quite fair”. As Referee #1 notes there are certainly some intriguing parallels between the cartilages we see in *Eriptychius* and gnathostome head development: we have adjusted the text to highlight the two of these that we think are most convincing based on the fossil evidence (the distinction between the splanchnocranium and neurocranium and the fact that only trabecular regions are mineralised, see below). We hope that this strikes a balance between what Referee #1 is looking for and the opinions of ourselves and Referees #2 and #3 and we hope and expect that this study will be integrated into future work on the evolution of the gnathostome head from a developmental perspective.

Some general comments:

Line 15: gnathostomes = *jawless ‘ostracoderms’ (stem gnathostomes) +jawed vertebrates. Maybe, crown gnathostomes is better*

Here we are explicitly introducing the two living jawed vertebrate groups before we introduce their extinct relatives in the next sentence.

Line 26: coalescence of cartilages into a single neurocranial unit is a derived trait

Here, maybe the ‘skull’ is better than ‘neurocranial unit’ because the coalescence of cartilages into a skull comprising the neurocranial cartilages and splanchnocranial cartilages.

We have changed neurocranial unit to skull here

Line 37-38 In living cyclostomes the cranial endoskeleton comprises an open trabecular structure holding the brain, and a feeding apparatus formed of a symmetrical set of paired and midline cartilages.

The trabecular structure in lampreys surrounds the hypophysis and extends anteriorly to the olfactory capsules. They are lateral rather than anterior to the notochord, and probably represent the anterior prolongation of parachordal cartilage. The developmental data indicate that the ‘lamprey trabecula’ develops from the mandibular mesoderm, and is not homologous with the gnathostome trabecula, which develops from premandibular neural crest cells.

Here we intended “trabecular” to mean “constructed from struts” rather than the developmental structures. In the context this was highly confusing phrasing: we have modified it to read “framework”.

Line 39-40 brain instead enclosed by a neurocranium, a single endoskeletal structure that encases the brain and nasal capsules, with paired mandibular and hyoid arches forming the feeding apparatus.

Yes, it is a major innovation that the trabeculae cranii and the visceral skeleton (mandibular and hyoid arches) are incorporated into cranial endoskeleton in gnathostomes.

This has been addressed by rewriting the text throughout to stress the incorporation of the visceral skeleton

Line 46-47 the plesiomorphic state for gnathostomes is a single, jawless neurocranial unit filling the space between the brain and dermal skeleton enclosing the brain and pharynx.

In galeaspids and osteostracans, the lateral visceral skeleton (splanchnocranium) merged with the middle braincase (neurocranium) to form a massive endoskeletal skull enclosing the brain and pharynx. Therefore, the endoskeleton filling the space between the brain and dermal skeleton includes the neurocranial unit+splanchnocranial unit. Maybe, 'endoskeletal unit' or 'skull' is better than 'neurocranial unit'.

We agree with this and have changed the text to address this (now line ~50).

Line 48-50 In the multiple other groups of Palaeozoic vertebrates all that is known of the cranial endoskeleton comes from difficult-to-interpret two-dimensional fossils or is inferred based on the dermal skeleton.

Yes, there are no evidence for the cranial endoskeleton in heterostracans. Janvier (1996) thought that the endoskeleton, whether cartilaginous or calcified, is probably absent in heterostracans because the internal surface of the armor plates is generally quite smooth and shiny, with a few nutrient canals, even at places marked with internal impressions of organs. In addition, the fact that the brain and labyrinth could leave such an accurate impression on the dermal skeleton suggests that they were enclosed only in fibrous sheath probably without intervening cartilage.

We agree, but think that this is included in the summary we give. The idea that Referee #1 refers to is included in the discussion, where we consider whether the posterior part of the skull in *Eriptychius* is not present or just missing.

Line 54: Sandbian (458.4-454.0 Ma) should be (458.4-453.0 Ma)

Well spotted, thanks! We have changed this (now line ~59).

Line 60-62: The cranial cartilages of *Eriptychius* are anatomically dissimilar from the crania of both cyclostomes and gnathostomes, revealing the existence of a third way of skeletonising the vertebrate cranium.

Maybe, it does not reveal the existence of a third way of skeletonising the vertebrate cranium, but represents a very, very important intermediate state before galeaspid and osteostracans, in which the gnathostome independent trabeculae cranii has formed, but the visceral skeleton (splanchnocranium) has not fused with the neurocranium to form a single massive endoskeletal skull. The fusion of the derivative structures of trabeculae and the visceral skeleton (splanchnocranium) with braincase happened later in galeaspids and osteostracans. See the comments for Line 125-150 for more details.

As described in other parts of the reply we would like to avoid direct interpretation of these cartilages in a developmental context. However, we agree that it is beneficial to emphasise the splanchnocranium vs neurocranium in the text and have rewritten to do this. We also think that describing it as “a third way” is probably oversimplistic. To address this we have modified the text in the summary paragraph and the introduction.

Line 73: *Eriptychius americanus* belongs to the stem-gnathostomata, not Agnatha any more

“Agnatha” is a formal group that is paraphyletic (if it includes Palaeozoic jawless fishes). “Stem-gnathostomata” is a misnomer as stem-groups are inherently paraphyletic and so can't be a formal group. We describe *Eriptychius* as an “agnathan”, i.e. as a descriptive term for a jawless fish rather than a formal grouping.

Line 74,93.....: The head dermal bone maybe use ‘plates’ better than ‘scales’

We agree with Referee #1 that “plates” is clearer when referring to dermal structures on the head as opposed to the body and have changed it in all text and figures.

Line 125-150: About the interpretation for the paired orbital cartilages, three paired cartilages symmetrical across the midline – two dorsal (termed mediolateral A, B) and one ventral (termed mediolateral C), and the median ventral cartilage.

It is very interesting that the authors have identified four paired cartilages symmetrical across the midline and the median ventral cartilage, which are distinct from that of known cyclostomes and gnathostomes, and probably represent a third way of constructing the vertebrate cranial endoskeleton. From my view, these structures may exactly compare to the derivative structures of trabeculae in galeaspids and gnathostomes. In crown

gnathostomes, the trabeculae derivative structures (e.g. orbitonasal lamina, postnasal wall, ethmoid process, interorbital septum, suborbital shelf, supraorbital crest, and rostrum) are fused with the anterior part of neurocranium (chordal cranium) to form the trabeculae cranii (prechordal cranium). In crown gnathostomes, the trabeculae cranii and derivative structures are of neural crest origin, developed mainly from the premandibular ectomesenchyme (Kuratani et al. 2001). However, the corresponding premandibular crest cells in lampreys developed into the upper lip and the floor of the nostril, or the nasohypophysial duct (Kuratani et al. 2001; Kuratani 2004). Therefore, they should bear a much more important evolutionary significance for our understanding the origin of gnathostome trabeculae cranii.

1) the paired orbital cartilages may compare to the cartilages of supraorbital crest in galeaspids and chondrichthyans.

In chondrichthyans, the supraorbital crest is an arched horizontal plate that extends posteriorly along the dorsal edge of the medial orbital wall from the preorbital process to the postorbital process, with its base continuous with the orbital wall and cranial roof (Compagno 1999). The supraorbital crest is penetrated medially by a row of supraorbital foramina for the tiny branches of the superficial ophthalmic nerve supplying the supraorbital lateral line canal (de Beer 1937). The supraorbital crest and the dorsal edge of the interorbital wall are delimited by the row of supraorbital foramina (de Beer 1937). Like crown gnathostomes, the supraorbital crest with a row of supraorbital foramina has been clearly identified in galeaspids. By contrast, the supraorbital crest is lacking in hagfishes and lampreys. The embryonic development of crown gnathostomes shows that the supraorbital crest is formed in the last stage (stage 6, 90mm, the skull has taken on the appearance of that of the adult) (de Beer 1937).

2) the paired mediolateral cartilages A may compared to the cartilages of the medial orbital wall (the interorbital septum) in galeaspids and chondrichthyans.

The medial wall of orbital cavity or the interorbital septum has been identified in galeaspids and chondrichthyans. In both groups, the medial orbital wall is delimited from the supraorbital crest by the row of supraorbital foramina. They are penetrated by numerous canals for the cranial nerves and vascular canals which show a similar arrangement. As the derivatives of trabeculae cranii (innovation of gnathostomes), the medial orbital walls are absent in hagfishes and lampreys.

3) the mediolateral cartilages C may compare to the cartilages of the suborbital shelf in galeaspids and chondrichthyans

In most chondrichthyans, the orbital cavity is bordered ventrally by the suborbital shelf, which tapers abruptly toward the midline to join its counterpart in a blunt point below the rostrum (Schaeffer 1981; Compagno 1999). The orbital cavity is also lined ventrally by a broad suborbital shelf in placoderms (Goujet 1984a). By contrast, the braincase of osteichthyans and fossil acanthodians is rather narrow and deep, with ventrally open orbital cavities without a suborbital shelf (Janvier 1996a). Like chondrichthyans and

placoderms, the orbital cavity of galeaspids is lined ventrally by a broad cartilaginous lamina for the suborbital shelf which separated the orbital cavity from the oralbranchial chamber completely.

4) the mediolateral cartilages B in *Eriptychius* may compare to the cartilages of orbitonasal lamina in galeaspids and chondrichthyans (not sure) .

In chondrichthyans, the trabeculae cranii grow forward into the rostral region to form the orbitonasal lamina, which surrounds the nasal sacs on each side, as well as the rostral cartilages (Jollie 1971). This structure lies between the anterior eye margin and nasal sac and forms the hind wall and part of the side wall of the nasal capsule (de Beer and Moy-Thomas 1935; de Beer 1937). The position of the orbitonasal lamina is also marked by the foramina for the profundus nerve (V1.f, Fig. 5.7A) and the so-called orbitonasal canal through which the orbitonasal vein (anterior facial vein) leaves the orbital cavity rostrally before entering the nasal capsule. de Beer (1937) pointed out that the existence of the ectethmoid chamber and of the orbitonasal foramen is evidence of the fact that the orbitonasal lamina is not an original part of the side wall of the cranial cavity, but is the original posterolateral wall of the nasal capsule, which has secondarily contacted with the wall of the cranial cavity. The orbitonasal lamina has been identified in galeaspids because the nasal capsules are located lateral to the base of the rostrum and rostral to the orbital cavity. Furthermore, the foramina for the profundus nerve, the so-called orbitonasal canal, and the canal for orbitonasal artery have also been identified in galeaspids. These foramina and canals in the orbitonasal lamina are almost the same as seen in gnathostomes. Thus, coupled with the position of the nasal capsules, the orbitonasal lamina is present in galeaspids. As the derivatives of trabeculae cranii (an innovation of gnathostomes), the orbitonasal lamina has never been identified in other jawless vertebrates because the nasal capsule of hagfishes, lampreys, and osteostracans is located high on the dorsal surface of the head, it is impossible to form a lamina between the orbital cavity and nasal capsules. Therefore, the orbitonasal lamina is lacking in hagfishes, lampreys, and osteostracans. In *Eriptychius*, the paired nasal capsules is probably extremely terminal and located in a space between the paired mediolateral cartilages A and the median ventral cartilage. The paired mediolateral cartilages B in *Eriptychius* is small and short, and maybe can be regarded as the precursor of the orbitonasal lamina.

5) The median ventral cartilage may compare to the cartilage of ethmoid plate (process) in galeaspids and chondrichthyans

In galeaspids, the orbitonasal lamina protrudes forward as a median cartilaginous rod separating and partly floored the paired olfactory bulbs. As the median cartilaginous rod is positioned just rostral to the hypophysial opening and ventral to forebrain, it is most readily compared to the ethmoid plate of chondrichthyans. In chondrichthyans, the paired trabeculae fuse with one another ventral to the brain, and project forward to form the rostral carina or ethmoid plate supporting the forebrain and rostrum ventrally (de Beer 1937; Wake 1979; De Iuliis and Pulerà 2007). The ethmoid plate (et.p, Fig. 5.7F) further

forms a vertical internasal septum between the nasal capsules (Wake 1979). However, compared to chondrichthyans, the ethmoid plate of galeaspids only supported the forebrain ventrally, but does not extend forward enough to form the internasal septum separating the paired nasal capsules. Judging from the position of olfactory bulb, the median ventral cartilage in Eriptychius is most likely corresponding to the ethmoid plate (process) in galeaspids and chondrichthyans.

We appreciate Referee #1 taking the time to make this detailed comparison. We agree that our manuscript will feed important data into understanding the evolution of the gnathostome head, and Referee #1 draws some interesting comparisons. These are based on positional relationships of cartilages associated with the trabeculae cranii in extant gnathostomes and identified in galeaspids. Of their suggestions, the identification of the paired orbital cartilages in *Eriptychius* as the supraorbital crest, and the median ventral cartilage as the ventral ethmoid plate are perhaps most plausible; in the former case the cartilage is associated with a concavity that can be identified as the orbit. In the latter, the ethmoid plate is associated with what we interpret as the olfactory bulbs, and we have identified a space for these on the ventral surfaces of paired cartilage A (Figure 2g), which are proximal to the median ventral cartilage (Figure 2c).

However, we are uncertain about some of the other proposed identifications, which depend on the presence of foramina and canals (cartilages A= interorbital septum; cartilages B=orbitonasal septum). We identified numerous canals within these cartilages (Figure 2d), and have modified the text to include the possibility that these are related to the superficial ophthalmic nerve (see comment below) but none of these show an arrangement that might be expected if they were associated with a row of foramina. Moreover *Eriptychius* is separated by a large time gap from anything where we can examine cranial development, and drawing direct identities for the cartilages in its head from the development of living gnathostomes is based on lots of assumptions, which can be problematic, as outlined above. It is instead our position that we should describe what we see, with minimal interpretation at this point, and then we hope other researchers incorporate it into their work on the evolution of development. As part of this cautious interpretation, as explained in other comments, we have elaborated on the comparison that we find most convincing: that the cartilages seem likely to be trabecular cranii derivatives due to their position, and that there is no evidence for any splanchnocranial contribution.

Line 131-133 : The anteriormost branchial scale lies postero-lateral to the other orbital cartilage (Figs 1, S3), putting the orbits and the first branchial arch in the same relative positions as other jawless stem-gnathostomes and cyclostomes.*T*

This indicates that the branchial arches of Eriptychius are independent, and are not incorporated into the neurocranium to form a massive skull as in osteostracans and galeaspids.

We agree, and have added additional text to clarify this point (now line ~122)

Line 152-155: All cartilages are pervasively penetrated by canals (Fig. 2g, S6). In the larger cartilages, i.e. the orbital cartilages and the mediolateral cartilages A, this tends to follow the pattern of a larger trunk entering the cartilage from the posterior side before splitting into smaller branches that open to the surface.

Maybe, these canals can be interpreted as the tiny branches of the superficial ophthalmic nerve supplying the supraorbital lateral line canal (de Beer 1937), which supporting the interpretation of the orbital cartilages and the mediolateral cartilages A as supraorbital crest and the medial orbital wall respectively.

We agree that a possible interpretation of the canals is as having carried nerves for sensory purposes. This is already discussed in the text. For the canals in the orbital cartilages the superficial ophthalmic nerve is a plausible candidate given their location and seeming origin in the front of the orbit (although its exits are not organised into a row, as noted above). We have modified the text to highlight the superficial ophthalmic nerve as a plausible identity for this canal, but as discussed we are reluctant to assign specific identities to the cartilages (now line ~148).

Line 203-207 The cranial cartilages of Eriptychius are not obvious homologues of endoskeletal structures in the head of any known extant or extinct vertebrate (Figs 2, S4).

The cranial cartilages of Eriptychius are probably homologues of the derivative structures of gnathostome trabeculae cranii. This indicates that the gnathostome trabeculae cranii arose as early as the Sandbian (458.4-454.0 Ma). This corroborated that the derivative structures of gnathostome trabeculae cranii are primitively independent as indicated by the development of gnathostomes. The coalescence of cranial cartilages and visceral arches into a single skull is a derived trait in galeaspid, and osteostracans, and jawed vertebrates.

(now line ~179) We have modified the text throughout to emphasise the comparison between the preserved cartilages and the trabeculae cranii, and the incorporation of the splanchnocranium into the skull in osteostracans and galeaspid. We note that Reviewer #3 specifically describes our statement heret as “fair”.

Line 220-222 This essentially appears to be the role of the cartilages in Eriptychius, although, unlike osteostracans and galeaspid, there is no evidence for any mineralisation from the level of the orbits posteriorly.

In galeaspid, the dorsal portion of branchial arches are incorporated into the neurocranium to form a massive skull (Gai et al, 2022) as assumed in osteostracans by Stensiö (1927, 1964). It has been known that the mandibular arch is located directly behind the orbital cavity and formed the posterior wall of orbital cavity. According to the authors, the anteriormost branchial scale lies postero-lateral to the other orbital cartilage (Figs 1,

S3), putting the orbits and the first branchial arch (mandibular arch) in the same relative positions as other jawless stem-gnathostomes. This indicates that the branchial arches of *Eriptychius* are independent, and are not incorporated into the neurocranium to form a massive skull as in osteostracans and galeaspids. This may explain why the orbits of *Eriptychius* remain unclosed posteriorly and no evidence for any mineralisation from the level of the orbits posteriorly.

We have modified the text to emphasise the incorporation of the splanchnocranium into the skull in osteostracans and galeaspids and the evidence that this is not the case in *Eriptychius*

Line 269 Figure 3C, Supplementary information Character 12, Character 38

A paired 'pineal' opening has been reported in the arandaspids *Arandaspis* (Ritchie & Gilbert-Tomlinson 1977) and *Sacabambaspis* (Gagnier 1993). However, the paired openings lie well behind the eyes, which are an unusual position for such structures. In addition, the paired openings are arranged in bilateral symmetry, which is very similar the condition of the paired endolymphatic openings in galeaspids, osteostracans, placoderms, and chondrichthyans on the dorsal surface of the head. So it is more likely that the paired 'pineal' openings in *Sacabambaspis* and *Arandaspis* are the paired endolymphatic openings. The real pineal opening of *Arandaspis* is a single opening and probably located in the single median pineal plate between the two eyes since the orbits are placed at the extreme anterior margin of the headshield in *Arandaspis* and *Sacabambaspis*. In lampreys, arandaspids, some heterostracans, anaspids, osteostracans, galeaspids, and placoderms, the pineal opening is a single structure perforating the dermal skull roof.

We hope that our work on *Eriptychius* causes other workers to revisit interpretations of the anatomy of other Ordovician vertebrate taxa. However as we do not add any new information on the anatomy of arandaspids we believe that this is beyond the scope of our study. As such we rely on the published interpretation of their anatomy.

Referee #2 (Remarks to the Author):

This manuscript provides the first 3D description of a long enigmatic skull of the Ordovician vertebrate Eriptychius, from the famous Harding Sandstone Formation of Colorado (USA), which yields some of the earliest vertebrates known to date. This specimen, first recorded by Denison in 1967, is an assemblage of elements of calcified cartilage pervaded by blood vascular canals, and odontode-bearing dermal bones, which was extremely difficult to interpret. It is now possible thanks to CT scanning which allows to make sense out of this crushed mix of bones and cartilage. Eriptychius, like most other vertebrates from the Harding Sandstone, was classically regarded as an “ostracoderm”, that is, a jawless stem gnathostome, and gathered with heterostracans in the pteraspidomorphs, based on the reputedly-yet debated- acellular structure of the dermal bone. This detail description of the first vertebrate endoskeleton, although somewhat puzzling for those who keep in mind the braincase structure of e.g. Silurian-Devonian or extant vertebrates, provides an unprecedented source of documentation for reconstructing vertebrate phylogeny, which the authors have attempted here masterly, and with insightful views on character transformations in early vertebrate history.

Many thanks to Referee #2 for their comments

Referee #3 (Remarks to the Author):

The moment I saw the figures I immediately grasped the importance of this paper. That the specimen was spotted (by Denison) and reconstructed to this detail (by the authors) is just as stunning as that it exists. I myself pulled hundreds of drawers containing blocks of Harding Sandstone sclerites, nonchalantly – and almost jokingly – scanning the surfaces for a surprise find. “Another Eriptychius graveyard,” I’d tell myself and move on after a few seconds of optical scanning. In fact, I’ve done that just a few days before receiving this manuscript. Now I’m sure that somewhere there I must have missed specimens like this under my nose. Nicely scored!

I sustain the major points from the authors as well justified. These are:

- The earliest three-dimensionally preserved vertebrate endocranium
- Independent cartilages forming the anterior part of that endocranium, **distinct from major patterns known in other vertebrate lineages**
- Dimensions and configurations of these skeletal parts support, and to some extent reflect, primitive patterns of the vertebrate skull as implied by recent phylogenetic shakeups around the vertebrate crown node.

To enhance this paper, I offer several major suggestions for the authors to consider, and a long list of minor comments that follows.

We thank Referee #3 for their detailed comments. We have addressed their minor and major points below.

1) The authors have an unparalleled opportunity here to address important predictions of the **New Head Hypothesis** (DOI: 10.1126/science.220.4594.268) and the **Heterotopy Hypothesis** (DOI: 10.1111/j.1525-142X.2011.00523.x), the former in the context of vertebrate origins and the latter in gnathostome evolution.

The New Head Hypothesis predicts that the prechordal cranium (an extensive forebrain and its skeletal capsule) **is a vertebrate novelty with the origin of neural crest**. Here, as the authors point out, the prechordal cranium occupies minimal dimensions in this stem gnathostome as in Cambrian stem vertebrates and Devonian stem cyclostomes (euphaneropids), suggesting that: a) the **"new head" was diminutive at best at the origin** of the vertebrate crown; and b) the extensive prechordal crania evolved independently in cyclostomes and gnathostomes. A diminutive prechordal cranium in Cambrian stem vertebrates has been considered preliminarily in the context of neural crest origins (DOI: DOI: 10.1126/science.adc9198; DOI: 10.1017/9781316832172.003).

The Heterotopy Hypothesis predicts that the origins of paired nasal capsules and trabecula cranii were linked, and were each a necessary condition for the origin of jaw skeleton. But here, *Eryptichius* appears to have paired nasal capsules (like *Haikouichthys*) and no discernible trabecula cranii or jaws. Might this imply that a) **paired nasal capsules are the ancestral for the vertebrate crown**; and b) the monorhiny condition independently evolved in cyclostomes and jawless stem gnathostomes (therefore cyclostomes, just like the case of filter-feeding larval lampreys, fail once again to serve as a surrogate ancestor with respect to gnathostomes)?

We agree with Referee #3 that this study feeds into hypotheses about the origins of the gnathostome head. However, we think some of the specific points they make here are based on a misreading of the manuscript.

New Head Hypothesis: While *Eriptychius* certainly has a short forebrain in comparison with modern mandibulate gnathostomes we were trying to make a comparison between it and *Sacabambaspis*, *Haikouichthys* etc in the final paragraph. Those other taxa have eyes directly at the front of the head, whereas based on our interpretation *Eriptychius* obviously has some part of the brain protruding between them, and substantial mineralisation in front of them. We agree that this new data feeds usefully into the NHH.

The Heterotropy Hypothesis: We suggest not that the paired lobes between the cartilages are nasal sacs, but rather that they are spaces for the olfactory bulbs (i.e. part of the forebrain). To our knowledge these are paired in all vertebrates, including those without paired nasal sacs such as lampreys, osteostracans, and hagfishes¹²³. So while this probably gives us some information about the likely location of the nasal sacs it is not possible to use it to distinguish between paired and unpaired nasal sacs. Because of this we don't think this is informative for the HH. Notably Referee #3 directly compares the structures we describe to the trabeculae cranii.

We have adjusted these parts of the paper to try and make these points clearer to the reader. We have also adjusted figure 3 to emphasise the preorbital head. We have cited these hypotheses in the introduction and touch upon the New Head hypothesis in the final paragraph of the discussion.

2) I believe the authors can do more to emphasize their closing (reducing) the temporal gap of the vertebrate braincase record. *Haikouichthys* (DOI: 10.1038/nature01264) – taken at the face value – represents the earliest 'vertebrate' with its neurocranium preserved. The next one up has so far been the galeaspid *Shuyu* from the Middle Silurian. So there has been nearly a 100-my gap of this information across when all major vertebrate lineages arose up to osteichthyan crown. It's amazing to have any endocranial information with one of the earliest vertebrates with mineralized skeletons.

We agree that this is an important point to make and have incorporated this into the introduction and discussion.

¹ Janvier, P. Les Céphalaspides du Spitsberg. Anatomie, phylogénie et systématique des Ostéostracés siluro-dévonien. Révision des Ostéostracés de la Formation de Wood Bay (Dévonien inférieur du Spitsberg). *Cahiers de Paléontologie, Section Vertèbres, Centre National de la Recherche Scientifique* (1985).

² Marinelli, W. & Strenger, A. *Vergleichende anatomie und morphologie der wirbeltiere. II. Lieferung.* Myxine glutinosa. (Deuticke, 1956).

³ Marinelli, W. & Strenger, A. *Vergleichende anatomie und morphologie der wirbeltiere. I. Lieferung.* Lampetra fluviatilis. (Deuticke, 1954).

3) Although it doesn't take much more to convince me of their anatomical reconstruction, I believe it would benefit the paper to strengthen their argument that the mineralized chondrocranial elements of *Eriptychius* truly forms the front part of the braincase. With the position of the proposed pineal foramen, and with the reconstruction of the olfactory bulbs and eyes, I have some difficult time wrapping my head around. I understand the challenging nature of this, but can the authors find additional evidence for the orbital identity (e.g., dermal scales preserved around the "orbital cartilages"; elaborating possible muscular attachment on orbital cartilage)? Any inference for nasal openings (e.g., penetration or demarcating on the paired fossae for "olfactory bulbs; break in dermal scales)? Notochordal sheath?

We don't have any additional material beyond what we describe. There is no evidence for a notochordal sheath: we would anticipate this to be found to the posterior of the endocranial elements that are mineralised in this specimen. Nor for nasal openings that we can see although this may in part due to be the collapse of the skeleton. There are orbital plates close to the inferred location of the displaced Right orbital cartilage, which we include in the text as part of our identification of the orbit and figure in Fig. 1 and extended data figure 4. These are small curved plates preserved in a chain, presumably pulled posteriorly with the displacement of the orbital cartilage (extended data figure 4c-e). These and the branchial plates provide anatomical constraints from the dermal skeleton on what the cartilages could comprise. I.e. they must lie in front of the branchial chamber (cf. comments from reviewer 1 regarding the separation of the branchial arches from the neurocranium) and lateral to the orbit.

We appreciate that this type of material is challenging to interpret and are open to alternative suggestions from other workers once the manuscript is published. However given that we are confidently able to identify the animal as a vertebrate, and the anatomical constraints that the dermal skeleton provides it is difficult to think of a part of the anatomy of a vertebrate that these cartilages could comprise other than the front of the neurocranium. As we touch upon elsewhere in the manuscript this actually matches up quite well with what one would expect to see using heterostracans as a rough model in that the posterior part of the head may be chiefly supported by the dermal skeleton.

I'm comfortable with the terminal mouth, and the authors made a good argument for this. Still, my first thoughts for the medial ventral cartilage were along the lines of "What if this is something like a hagfish's subnasal cartilage, only pressed down to the ventral shield during fossilization?"

The median ventral cartilage is closely associated with the underlying squamation (including the “rostral scales”), which fits around it closely. This can be seen in figure 1 and in the extended data figure 3. Thus we infer that it overlay the ventral squamation directly. Moreover, it is preserved underlying the other cartilages: to be a subnasal cartilage equivalent it would have to have been pushed posteriorly to its position below them while remaining “lined up” with them and not disarticulating the patch of squamation below it. We consider this unlikely (e.g. compare to the posteriorly displaced orbital cartilage, which has travelled posteriorly but has been disassociated from any articulated dermal squamation presumably as part of this post-mortem dislocation

Because the endo- and exoskeletons are loosely connected, it seems to me that we can only be sure of their dorsoventral positions relative to each other, but not in absolute sense between the dorsal and ventral part of its head. There are free paired cartilages in both hagfish and lamprey heads in this region. Granted, these cartilages in *Eryptichius* look like none of those, and I’m not suggesting they correspond. What I would love to see here instead is evidence for the interaction between these cartilages and the nervous system. Much to ask, yes, but this would really strengthen their argument.

We think we have addressed the nervous system to the best of our abilities in the two replies above, re: possible relationship of the cranial canals to the superficial ophthalmic nerve. The orientation of the concavity suggested to represent the orbit in Figure 1c, d supports our interpretation of dorsal and ventral for the specimen as a whole. Another piece of evidence is provided by our interpretation of the position of the olfactory bulbs and the pineal in Figure 2, dorsal structures.

4) As an addendum to my points above, I have concerns over the terminologies such as “cranial endoskeleton” and “trabecular”. As for the former, I appreciate that the authors are trying to navigate carefully around the kind of problem I’m just about to point out on “trabecular”. However, their internal definition is unclear and I wonder if it’s used softly in one place and rigidly in another. They should either: a) define at the outset what they mean by “cranial endoskeleton” and stick to that definition; or b) use more conventional, clearly defined terms such as chondrocranial elements, prechordal cranium, neurocranium, etc. As for “trabecular”, in a chondrocranium this element derives from the ectomesenchyme in a prechordal position. Therefore, as the authors acknowledge, “trabecula” in cyclostomes are parachordal elements. Controlling this terminology will of course be important to address their findings in the context of the New Head and Heterotopy hypotheses.

As Referee #3 observes we were attempting to navigate around what we saw as various terminological problems. However we agree with the reviewer that this is an important thing to tighten up and have addressed this throughout the text. In particular we have replaced “cranial

endoskeleton” with “neurocranium” where applicable (contra Reviewer 1. Also as noted above we have clarified our use of the term ‘trabecular’ especially with the confusing use in the introduction).

5) I think it would be important to clearly illustrate discrepancies with respect to Denison’s observation. This may be done in a supplementary figure, where Denison’s reconstruction is presented side by side with the new scheme.

We do this in Extended data figure 2, which shows the specimen labelled with Denison’s interpretations of elements (in italics) vs our interpretations. For clarity, Denison only provided annotations of the specimen rather than attempt a reconstruction (in figure 2 of his description of PF 1795).

MINOR COMMENTS

Line 14: “... its form in the two living vertebrate groups...”

“... its form between the two living vertebrate groups...”

We have made this change.

Lines 16-18: “the earliest three-dimensionally preserved vertebrate cranial endoskeletons are from Silurian-Devonian taxa that do little to resolve modern states.”

Here, I think it’s better to focus on the numbers. E.g.: Previously, no three-dimensionally preserved neurocranium has been known from the first 100 my of vertebrate evolutionary history.

We have rewritten this sentence to try and do this

Line 19: “...endoskeletal cranial anatomy...”

This is somehow awkward. Can it not be endocranial anatomy?

We have made this change.

Line 25: “...more akin to crownward taxa...”

The authors mean “crownward gnathostomes” here, correct?

We have addressed this by explicitly listing the taxa instead.

Line 30: “...existed in the earliest skeletonising jawed vertebrates.”

I’m not sure if this is correct. Perhaps “existed among the earliest known stem gnathostomes”?

We have modified the text to address this.

Line 35: I’m not sure if “frustrated” is the word here. Perhaps “challenged”?

We have changed this to hampered

Line 37 and elsewhere: Watch out for the homology of trabecula cranii whenever this structure in cyclostome is discussed. The authors understand this, but not all readers do.

We have addressed this in a reply to Referee #1.

Line 38: THE feeding apparatus CONSISTING of a...

We have made this change.

Lines 40-41: “...a single endoskeletal...capsules...”

Put this in parentheses.

This is no longer relevant due to rewriting

Lines 42-44: “...to polarise these conditions the fossil record of stem-group gnathostomes has proved key to reconstructing their evolution, but does little to bridge the morphological gap between cyclostomes and gnathostomes.”

This needs to be more precise (and less repetitive). “...to polarise these conditions, the fossil record of stem-group gnathostomes would be critical to reconstructing their evolution, but has yielded little to bridge the morphological gap.”

We have addressed this by modifying the text

Lines 45-46: "...the Silurian and Devonian placoderms, osteostracans, and galeaspids stem gnathostomes together..."

"...the Silurian and Devonian placoderms, osteostracans, and galeaspids together..."

"...the Silurian and Devonian stem gnathostomes together..."

"...the Silurian and Devonian stem gnathostomes (placoderms, osteostracans, and galeaspids) together..."

We have addressed this by modifying the text

Lines 47-48: Does a neurocranium "enclose" the pharynx? That would be splanchocranium.

Agreed. We have changed the text to address (see also response to Referee #1)

Lines 73-76: Emended diagnosis.

This first part of diagnosis is based on what would have been present in other Ordovician astraspids and eryptychiids but just happens to be preserved in this one taxon, so I'm not sure if this is appropriate for diagnosis of Eriptychius itself. This should refer to the histology of these elements (described in detail in Supplementary Text) and specific patterns discerned in the dermal skeleton of this taxon (which are mentioned toward the end of differential diagnosis).

We have rewritten the diagnosis to match the reviewers specifications

Line 118 "further" should be "farther"

We have made this change (now line ~106).

Lines 130-132: The relationship between the orbit and the first branchial arch varies among cyclostomes and stem gnathostomes (and has been described also in relation to the otic capsule, though that can only be distantly inferred here), so this needs some elaboration. How posterolateral?

It is difficult to be exact due to the collapse for the fossil: we have tried to clarify this in the text. (now line ~119-120).

Lines 152-166: Good addition and restrained comparison.

Line 203-214: I think this is quite fair.

Line 226: Please be clear about the “trabecular region” – do the authors mean a prechordal cranium here? Or something else?

We have clarified this by changing the terminology to explicitly state prechordal (now line ~201).

Line 238: Cyclostomes also have a single neurocranium, only that it forms from the parachordal component (paraxial mesoderm) and has little contribution from the prechordal head.

We have rewritten to make this clear: what we meant was a single enclosing neurocranium (now line ~214).

Lines 243-254: I understand that the authors are navigating carefully to drive two complex points here: 1) Eriptychius offers the earliest evidence of a preorbital skeleton (=prechordal cranium?); but 2) the inferred anterior position of the orbits implies that a small or diminutive preorbital head is a widespread and potentially primitive condition among early vertebrates. I think these two points can be more explicitly or clearly delineated by driving home the position of the orbit as a landmark to delimit the preorbital/prechordal region.

Related to our reply above to the major point 1 of Referee #3 we were trying to drive home a slightly different set of points. In either case emphasising the role of the position of the orbit is a good idea that we have tried to incorporate into the text and figure 3

Figure 3: panels a-e should be linked to the corresponding terminal taxa on the tree.

A generalised cyathaspid (d) is somewhat misleading because the grey shades (imprints on the dermal shield) cannot be distinguished from those for endoskeletal elements in the other taxa.

Similarly, the dermal skull is not clearly distinguished against the endocranial elements in the osteostracan and placoderm (f, g) – why this is unshaded in Samabambapis, cyathaspid, and Eriptychius but shaded in Benneviaspis and Dicksonosteus?

There should be some labels to distinguish nasal apertures (b, c, f, g) against capsules (a, d, e?) because they occupy different positions.

Pineal (yellow) – colour code is difficult to make out.

We have modified Figure 3 to address these points by:

- Adding linking lines connecting taxa to their names
- Tweaking the colour scheme to emphasis dermal vs endoskeletal and keep consistency
- Tweaking the colour scheme to distinguish between the endocranial elements and the brain (with dotted lines to represent imprints in the cyathaspid)
- We have removed the nasal blobs from the images as they were confusing (as the referee points out) and unnecessary.
- We have changed the colour scheme to try and be clear: a yellow is still involved but is darker and on the larger eye-blobs.

Reviewer Reports on the First Revision:

Referees' comments:

Referee #1 (Remarks to the Author):

I am happy to see that the potential evolutionary significance for the origin of gnathostome trabeculae cranii has been addressed in the revised manuscript and agree with authors not imposing a rigid identity on the cartilages based on the development of modern groups, and just construct a prudent hypothesis that will be testable by future researchers. This study will be integrated into future work on the evolution of the gnathostome head from a developmental perspective. Most of the points raised in my previous round of review have been satisfactorily addressed except one minor question.

Line 269 Figure 3C, Supplementary information Character 12, Character 38

A paired 'pineal' opening has been reported in the arandaspids *Arandaspis* (Ritchie & Gilbert-Tomlinson 1977) and *Sacabambaspis* (Gagnier 1993). However, the paired openings lie well behind the eyes, which are an unusual position for such structures. In addition, the paired openings are arranged in bilateral symmetry, which is very similar the condition of the paired endolymphatic openings in galeaspids, osteostracans, placoderms, and chondrichthyans on the dorsal surface of the head. So it is more likely that the paired 'pineal' openings in *Sacabambaspis* and *Arandaspis* are the paired endolymphatic openings. The real pineal opening of *Arandaspis* is a single opening and probably located in the single median pineal plate between the two eyes since the orbits are placed at the extreme anterior margin of the headshield in *Arandaspis* and *Sacabambaspis*. In lampreys, arandaspids, some heterostracans, anaspids, osteostracans, galeaspids, and placoderms, the pineal opening is a single structure perforating the dermal skull roof.

We hope that our work on *Eriptychius* causes other workers to revisit interpretations of the anatomy of other Ordovician vertebrate taxa. However as we do not add any new information on the anatomy of arandaspids we believe that this is beyond the scope of our study. As such we rely on the published interpretation of their anatomy.

Comments:

The original interpretation for the paired 'pineal' opening in the arandaspids was obviously wrong for my previously listed reasons. One more evidence is that the paired openings in arandaspids is also located close to the median transverse canals as in galeaspids, osteostracans, and placoderms (see figure). The error has been corrected in our new paper "The first galeaspid fish (stem-gnathostomata) from the Silurian Xiushan Formation of Hunan Province, China" because a pair of similar pores are also well preserved in our new materials of galeaspids. Our paper has been accepted by the Historical Biology and will be online in next week or at the end of June, which will be available at the following permanent link:

<https://doi.org/10.1080/08912963.2023.2225083>.

I wish the authors can consider it and give a correction in their phylogeny and Figures.

After the minor question is addressed, I am happy to support its publication in Nature. I look forward to seeing it in print.

Referee #2 (Remarks to the Author):

The authors have provided extremely detailed and sensible responses to the reviewers, and modified their manuscript accordingly. This revised version is, to me, a masterpiece, which will certainly fuel important new insights on the evolution of the total-group gnathostome skull. I look forward to seeing this piece published, as much as I hope to see a braincase of an arandaspid!

Referee #3 (Remarks to the Author):

Nature ms 2023-04-05562A “The oldest three-dimensionally preserved vertebrate neurocranium” presents a revision that in content improved from the original submitted version. I appreciate the thoughts and care with which the authors revised the manuscript, and firmly stand by my previous assessment about the importance of this paper. However, the revision ironically exposed parts of the narrative or of the writing that now seem to fit a bit uncomfortably. These issues can and should be fixed with not too much additional work.

Before I get into those points, I wish to bring their attention back to a few of my initial comments to which the authors, overall, responded quite reasonably.

First, my suggestion to highlight comparison and differences with the early description of the same specimen by Denison (1967). The authors responded that they already had them addressed in their extended data figure. Sure, by labels they did. However, Denison illustrated the specimen along with his interpretations (his Figure 2, Denison 1967). My recommendation was to republish these images with his interpretation and contrast it with the new interpretation.

Second, I appreciate that the authors took care of “cranial endoskeleton” in most parts of the manuscript. However, this term (or its reincarnation) still sticks around (Line 101, Line 188, Line 208, Line 228). Now without the proper context or definition, it is even less clear what this term refers to.

Line 59: “endoskeletal cartilage”

Is there a chondrocranial element that is not endoskeletal?

Line 120: “branchial endoskeletal elements”

Is there any branchial element that is not endoskeletal?

Line 23-24: “...the neurocranium filled and framed the head...”

Line 188: “... framing and filling out the head skeleton.”

Line 208: “...frames and fills the head...”

I think the description should be more precise. The authors did this nicely in Lines 34-38 so can it be

condensed to replace this expression? The cyclostome neurocranium (parachordal cartilages, nasohypophyseal cartilages, otic capsules, and interotic notochordal sheaths...) also “fills and frames” the head (unless the authors define this explicitly), whereas in the gnathostome chondrocranium, the neurocranium proper (‘endoskeletal neurocranium’ as the authors originally referred to) remains incomplete in many species, except in chondrichthyans. The sentence in question is also convoluted.

Lines 27-28: “...a hitherto unknown way of neurocranial construction...”

Line 64: “...a distinct way of skeletonising the vertebrate head...”

I’m not sure what “way” or “construction” biologically refers to.

Line 75: I’m not very clear on the condition where acellular bone is “surmounted” by ornament, which consists of “very” coarse tubular dentine. How coarse is very coarse?

Lines 108-110: This is difficult to crunch through.

Lines 120-121: “... the branchial arches were not incorporated into the skull...”

Branchial arches are by definition a part of a skull. A skull consists of dermatocranium, neurocranium, and splanchoocranium (which includes the branchial arches). What is meant when the branchial arches are “incorporated into the skull”?

Lines 150-154: This is a long sentence spanning over four lines in which the word “cartilage” is used six times. Nothing is wrong in this sentence grammatically or scientifically, and this is something caught at later stages I trust. However, it distracts readers from focusing on the science. Similar compositions are seen across the text, some of which I marked on the scanned manuscript.

Line 178: What is a single “cage of cartilaginous struts”? Yes, the authors explained this, but the “cage” terminology I am not sure about being in agreement.

Line 192: “Splanchoocranium” is a part of the skull. What does “incorporated” mean here?

Line 196-200: Another long, convoluted sentence that keenly needs editing. It is unclear whether the authors discuss historical or ontogenetic development – I interpret the latter, but it could still be read otherwise. It seems the authors have too much to say in this concluding sentence of the paragraph. Please provide some traffic control of ideas.

Lines 205-206: Within the same sentence, two tenses exist (“...recovers...” and “...resolved...”).

Lines 207-209: The verbs are not matching within this long sentence (“...would extend” and “...shows...”).

Line 212: “...these early gnathostome neurocrania calcified...”

Is there evidence for these elements being calcified *sensu stricto*? Do these correspond to the elements sectioned for Fig. 13 of Denison (1967)?

Lines 212-215: This long convoluted clause needs some editing. The way it's embedded in the paragraph, its significance is concealed. It is also not clear whether the authors are referring to historical or ontogenetic development – it is the latter, but could be read otherwise.

L215-216: The authors conclude this paragraph with the statement that the “complex evolutionary history [as implied by the discovery of the neurocranium of Eriptychius] ... explain why it has been so difficult to” directly compare cyclostome and gnathostome skulls. Is this the problem they wish to solve here? From the original version, I fully embrace the narrative that here we are looking at a unique neurocranium that doesn't fit cyclostome or gnathostome patterns – in that sense, what they should truly unpack and emphasize here is the clause over Lines 212-215.

Whether this resolves the cyclostome-gnathostome skull homology comes with several assumptions that are implicit here – for example, the argument would be valid if the condition in Eriptychius represents a symplesiomorphy for gnathostomes. However, this is one occurrence of such a pattern in the gnathostome stem; the phenotypic distance between cyclostomes and gnathostomes remains unchanged (=homology still cannot be established); Eriptychius does not exactly serve as an intermediate or reduce the gap (at least for homology of individual elements). So is Eriptychius helping anything here? I think this would require much longer discussion with supporting evidence.

Lines 220-221: “Based on the lateral positions of the orbits as well as the nasal openings, THIS [emphasis mine] seems likely...”

I think the context has to be clarified here that the authors are only referring to Astraspis and heterostracans, as the nasal openings cannot be determined in Eriptychius.

Line 219: “...a preorbital, perhaps prechordal...”

Preorbital assumes prechordal. Embryologically, the notochord terminates; anterior to that, the prechordal/premandibular cartilages form. Here, the polar cartilages and trabecular cranii sit posterior to ethmoid elements (=interorbital, preorbital, and nasal elements).

Lines 222-227: I appreciate that the authors considered review comments, and I overall agree with the authors' assessment. However, I think some sensitivity is required for this discussion. The orbital openings are not always a good landmark for delineating the preorbital/prechordal skeleton internally. For one good counter-example, see some heterostracans, such as Figure 4.5H in Janvier (1996) Early Vertebrates. In this particular case, the soft prechordal skull sits medially and posteriorly to the external orbital openings.

Line 229: Which “recent phylogenies”? ;)

Finally, I made scribbles on the manuscript pages about stylistic and compositional issues. I waived on them in the original submission as I trust that these things will be addressed properly down the road, but this time they became more conspicuous and did get in my way of reading after the authors took care of other, scientific comments. I scanned the pages and appended the pdf to my review comments. This is my preference/suggestion and for the authors to just consider. I am aware of my overstepping and definitely not charging the authors to revise every one of them.

Author Rebuttals to First Revision:

Referees' comments:

Referee #1 (Remarks to the Author):

I am happy to see that the potential evolutionary significance for the origin of gnathostome trabeculae cranii has been addressed in the revised manuscript and agree with authors not imposing a rigid identity on the cartilages based on the development of modern groups, and just construct a prudent hypothesis that will be testable by future researchers. This study will be integrated into future work on the evolution of the gnathostome head from a developmental perspective. Most of the points raised in my previous round of review have been satisfactorily addressed except one minor question.

Line 269 Figure 3C, Supplementary information Character 12, Character 38

A paired 'pineal' opening has been reported in the arandaspids *Arandaspis* (Ritchie & Gilbert-Tomlinson 1977) and *Sacabambaspis* (Gagnier 1993). However, the paired openings lie well behind the eyes, which are an unusual position for such structures. In addition, the paired openings are arranged in bilateral symmetry, which is very similar the condition of the paired endolymphatic openings in galeaspids, osteostracans, placoderms, and chondrichthyans on the dorsal surface of the head. So it is more likely that the paired 'pineal' openings in *Sacabambaspis* and *Arandaspis* are the paired endolymphatic openings. The real pineal opening of *Arandaspis* is a single opening and probably located in the single median pineal plate between the two eyes since the orbits are placed at the extreme anterior margin of the headshield in *Arandaspis* and *Sacabambaspis*. In lampreys, arandaspids, some heterostracans, anaspids, osteostracans, galeaspids, and placoderms, the pineal opening is a single structure perforating the dermal skull roof.

We hope that our work on *Eriptychius* causes other workers to revisit interpretations of the anatomy of other Ordovician vertebrate taxa. However as we do not add any new information on the anatomy of arandaspids we believe that this is beyond the scope of our study. As such we rely on the published interpretation of their anatomy.

Comments:

The original interpretation for the paired 'pineal' opening in the arandaspids was obviously wrong for my previously listed reasons. One more evidence is that the paired openings in arandaspids is also located close to the median transverse canals as in galeaspids, osteostracans, and placoderms (see figure). The error has been corrected in our new paper "The first galeaspid fish (stem-gnathostomata) from the Silurian Xiushan Formation of Hunan Province, China" because a pair of similar pores are also well preserved in our new materials of galeaspids. Our paper has been accepted by the Historical Biology and will be online in next week or at the end of June, which will be available at the following permanent link:

<https://doi.org/10.1080/08912963.2023.2225083>.

I wish the authors can consider it and give a correction in their phylogeny and Figures.

After the minor question is addressed, I am happy to support its publication in Nature. I look forward to seeing it in print.

We thank Referee #1 for their comment here and for providing the reference to their work. While this is an interesting proposition that has previously been mentioned in unpublished work, there are a number of significant problems (not least the implications for the prechordal structure in arandaspids) with it that we feel are beyond the scope of our manuscript. Given these and the timing of publication with respect to the current work we feel it is reasonable to stay with the interpretations of Gagnier and others and adopt these in our phylogenetic analysis. However, we have cited Referee #1's alternative interpretation in the figure caption for figure 3 and in the relevant phylogenetic characters in the SI.

Referee #2 (Remarks to the Author):

The authors have provided extremely detailed and sensible responses to the reviewers, and modified their manuscript accordingly. This revised version is, to me, a masterpiece, which will certainly fuel important new insights on the evolution of the total-group gnathostome skull. I look forward to seeing this piece published, as much as I hope to see a braincase of an arandaspid!

We are extremely grateful for the positive comments of Referee #2. We'd also like to see the braincase of an arandaspid and are, possibly not alone, looking for one.

Referee #3 (Remarks to the Author):

Nature ms 2023-04-05562A "The oldest three-dimensionally preserved vertebrate neurocranium" presents a revision that in content improved from the original submitted version. I appreciate the thoughts and care with which the authors revised the manuscript, and firmly stand by my previous assessment about the importance of this paper. However, the revision ironically exposed parts of the narrative or of the writing that now seem to fit a bit uncomfortably. These issues can and should be fixed with not too much additional work.

We thank the referee for their comments, we have attempted to address their remaining comments as detailed below.

Before I get into those points, I wish to bring their attention back to a few of my initial comments to which the authors, overall, responded quite reasonably.

First, my suggestion to highlight comparison and differences with the early description of the same specimen by Denison (1967). The authors responded that they already had them addressed in their extended data figure. Sure, by labels they did. However, Denison illustrated the specimen along with his interpretations (his Figure 2, Denison 1967). My recommendation was to republish these images with his interpretation and contrast it with the new interpretation.

Denison's 1967 Fieldiana paper is freely available online via the Biodiversity Heritage Library, and the italicised portions of our labels correspond directly to his labels: we have modified the caption of the figure to point the reader towards the specific figure. However due to the free availability of Denison's paper and the need to obtain image rights we have not included his illustration.

Second, I appreciate that the authors took care of "cranial endoskeleton" in most parts of the manuscript. However, this term (or its reincarnation) still sticks around (Line 101, Line 188, Line 208, Line 228). Now without the proper context or definition, it is even less clear what this term refers to.

We have rewritten these to incorporate clear terminology.

Line 59: "endoskeletal cartilage"

Is there a chondrocranial element that is not endoskeletal?

We have changed this to read "cartilages"

Line 120: "branchial endoskeletal elements"

Is there any branchial element that is not endoskeletal?

We have rewritten this section of the text to address this

Line 23-24: "...the neurocranium filled and framed the head..."

Line 188: "... framing and filling out the head skeleton."

Line 208: "...frames and fills the head..."

I think the description should be more precise. The authors did this nicely in Lines 34-38 so can it be condensed to replace this expression? The cyclostome neurocranium (parachordal cartilages, nasohypophyseal cartilages, otic capsules, and interotic notochordal sheaths...) also "fills and frames" the head (unless the authors define this explicitly), whereas in the gnathostome chondrocranium, the neurocranium proper ('endoskeletal neurocranium' as the authors originally referred to) remains incomplete in many species, except in chondrichthyans. The sentence in question is also convoluted.

We have rewritten these parts of the text to use precise terminology.

Lines 27-28: "...a hitherto unknown way of neurocranial construction..."

Line 64: "...a distinct way of skeletonising the vertebrate head..."

I'm not sure what "way" or "construction" biologically refers to.

We have rewritten this section as part of the shortening of the summary paragraph to clarify this point.

Line 75: I'm not very clear on the condition where acellular bone is "surmounted" by ornament, which consists of "very" coarse tubular dentine. How coarse is very coarse?

A distinctive feature of *Eriptychius* is the wide-calibre coarse tubules that form the ornament of the dermal armour. We have modified the text to clarify. .

Lines 108-110: This is difficult to crunch through.

We have rewritten this sentence for clarity.

Lines 120-121: "... the branchial arches were not incorporated into the skull..."

Branchial arches are by definition a part of a skull. A skull consists of dermatocranium, neurocranium, and splanchocranium (which includes the branchial arches). What is meant when the branchial arches are "incorporated into the skull"?

We meant into a single unit, but the reviewer is correct that we were unclear. We have rewritten this.

Lines 150-154: This is a long sentence spanning over four lines in which the word "cartilage" is used six times. Nothing is wrong in this sentence grammatically or scientifically, and this is something caught at later stages I trust. However, it distracts readers from focusing on the science. Similar compositions are seen across the text, some of which I marked on the scanned manuscript.

We have rewritten this sentence and split it into two. We have tried to modify other similar sentences.

Line 178: What is a single "cage of cartilaginous struts"? Yes, the authors explained this, but the "cage" terminology I am not sure about being in agreement.

We have rewritten this to use our terminology from the introduction (i.e. "open framework").

Line 192: "Splanchocranium" is a part of the skull. What does "incorporated" mean here?

We have rewritten this: as in one of the comments above we meant to say fused with the neurocranium into a single unit.

Line 196-200: Another long, convoluted sentence that keenly needs editing. It is unclear whether the authors discuss historical or ontogenetic development – I interpret the latter, but it could still be read otherwise. It seems the authors have too much to say in this concluding sentence of the paragraph. Please provide some traffic control of ideas.

We have rewritten the last two sentences of this paragraph to address the reviewers points here.

Lines 205-206: Within the same sentence, two tenses exist ("...recovers..." and "...resolved...").

We have removed the word "resolved"

Lines 207-209: The verbs are not matching within this long sentence ("...would extend" and "...shows...").

We have changed to “extend”

Line 212: “...these early gnathostome neurocrania calcified...”

Is there evidence for these elements being calcified *sensu stricto*? Do these correspond to the elements sectioned for Fig. 13 of Denison (1967)?

Globular calcified cartilage has been extensively described from the Harding Sandstone and lateral equivalents, including by Denison in Fig. 13, and this is how the cartilages in PF 1795 are interpreted by Denison. We follow this interpretation in the manuscript which we have made this clearer by adding “globular calcified cartilage” at line 93 of the description.

Lines 212-215: This long convoluted clause needs some editing. The way it’s embedded in the paragraph, its significance is concealed. It is also not clear whether the authors are referring to historical or ontogenetic development – it is the latter, but could be read otherwise.

We have rewritten this sentence and tweaked the paragraph as a whole a bit to make it clearer that we are talking about the phylogenetic implications. We have also added a citation in this new text, Donoghue et al 2000, which is the only previous phylogenetic analysis to include *Eriptychius* (recovering it as a stem-group gnathostome).

L215-216: The authors conclude this paragraph with the statement that the “complex evolutionary history [as implied by the discovery of the neurocranium of *Eriptychius*] ... explain why it has been so difficult to” directly compare cyclostome and gnathostome skulls. Is this the problem they wish to solve here? From the original version, I fully embrace the narrative that here we are looking at a unique neurocranium that doesn’t fit cyclostome or gnathostome patterns – in that sense, what they should truly unpack and emphasize here is the clause over Lines 212-215.

Whether this resolves the cyclostome-gnathostome skull homology comes with several assumptions that are implicit here – for example, the argument would be valid if the condition in *Eriptychius* represents a symplesiomorphy for gnathostomes. However, this is one occurrence of such a pattern in the gnathostome stem; the phenotypic distance between cyclostomes and gnathostomes remains unchanged (=homology still cannot be established); *Eriptychius* does not exactly serve as an intermediate or reduce the gap (at least for homology of individual elements). So is *Eriptychius* helping anything here? I think this would require much longer discussion with supporting evidence.

The point we were trying to make here was not that *Eriptychius* resolves homology between gnathostome and cyclostome states (we are agreed with the reviewer that it does not). But rather that the fact you see something so different to either in an Ordovician stem-gnathostome rather than something intermediate between cyclostomes and gnathostomes hammers home the point that the search for such homologies is probably futile. So as not to add anything substantial as requested by the editor we have limited ourselves to rewriting this sentence to try and make this point clearer.

Lines 220-221: “Based on the lateral positions of the orbits as well as the nasal openings, THIS

[emphasis mine] seems likely...”

I think the context has to be clarified here that the authors are only referring to Astraspis and heterostracans, as the nasal openings cannot be determined in Eriptychius.

We agree and have swapped around the clauses of this sentence, which makes the meaning clearer

Line 219: “...a preorbital, perhaps prechordal...”

Preorbital assumes prechordal. Embryologically, the notochord terminates; anterior to that, the prechordal/premandibular cartilages form. Here, the polar cartilages and trabecular crania sit posterior to ethmoid elements (=interorbital, preorbital, and nasal elements).

We have rewritten this to just state “prechordal”

Lines 222-227: I appreciate that the authors considered review comments, and I overall agree with the authors’ assessment. However, I think some sensitivity is required for this discussion. The orbital openings are not always a good landmark for delineating the preorbital/prechordal skeleton internally. For one good counter-example, see some heterostracans, such as Figure 4.5H in Janvier (1996) Early Vertebrates. In this particular case, the soft prechordal skull sits medially and posteriorly to the external orbital openings.

We think we give the point reasonable discussion, given the short-form of the article. The cited example is a speculative reconstruction of the brain in an amphiaspid, an extremely anatomically unusual, geographically and temporally limited group of Early Devonian heterostracans. While we agree these animals raise interesting questions about the relative placement of the eyes/brain in early vertebrates we think that it is too phylogenetically and temporally separated from the Ordovician taxa under discussion to be relevant within the scope of the manuscript.

Line 229: Which “recent phylogenies”? ;)

We have added citations!

Finally, I made scribbles on the manuscript pages about stylistic and compositional issues. I waived on them in the original submission as I trust that these things will be addressed properly down the road, but this time they became more conspicuous and did get in my way of reading after the authors took care of other, scientific comments. I scanned the pages and appended the pdf to my review comments. This is my preference/suggestion and for the authors to just consider. I am aware of my overstepping and definitely not charging the authors to revise every one of them.

Thanks for this, we have looked through the manuscript pages and tried to tweak cumbersome sentences etc.